# Retrospective model-based inference guides model-free credit assignment

Rani Moran [1,2], Mehdi Keramati[1,2,3], Peter Dayan [1,4,5] & Raymond J. Dolan [1,2]

An extensive reinforcement learning literature shows that organisms assign credit efficiently, even under conditions of state uncertainty. However, little is known about credit-assignment when state uncertainty is subsequently resolved. Here, we address this problem within the framework of an interaction between model-free (MF) and model-based (MB) control systems. We present and support experimentally a theory of MB retrospective-inference. Within this framework, a MB system resolves uncertainty that prevailed when actions were taken thus guiding an MF credit-assignment. Using a task in which there was initial uncertainty about the lotteries that were chosen, we found that when participants' momentary uncertainty about which lottery had generated an outcome was resolved by provision of subsequent information, participants preferentially assigned credit within a MF system to the lottery they retrospectively inferred was responsible for this outcome. These findings extend our knowledge about the range of MB functions and the scope of system interactions.

[1] Max Planck UCL Centre for Computational Psychiatry and Ageing Research, University College London, 10-12 Russell Square, London WC1B 5EH, UK. [2] Wellcome Centre for Human Neuroimaging, University College London, London WC1N 3BG, United Kingdom. [3] Department of Psychology, City, University of London, London EC1R 0JD, UK. [4] Gatsby Computational Neuroscience Unit, University College London, London W1T 4JG, UK. [5] Max Planck Institute for Biological Cybernetics, Max Plank-Ring 8, 72076 Tuebingen, Germany. Correspondence and requests for materials should be addressed to R.M. (email: rani.moran@gmail.com)

fficient adaptation to the environment requires that organisms solve a credit-assignment problem (i.e. learn which actions are rewarding in different world-states). Previous research has demonstrated that organisms use flexible efficient learning strategies to cope with situations that entail uncertainty about the state of the world[1–7]. However, little attention has been paid to a common case where there is uncertainty about a state at the time an action is executed and an outcome is received, but where this state uncertainty can subsequently be resolved by an inference process. This retrospective resolution can dramatically color and explain (away) the implications of action-outcome pairings. Indeed, whole genres of detective fiction depend on this very scenario, as does the dawning realisation of an unwitting victim fleeced by a devious card shark, who had initially seduced the victim into thinking they are skilled or blessed with good luck by providing early rewards. The question we address here concerns the effect of this retrospective inference on credit-assignment and whether, and how, it modulates fundamental signatures of reinforcement learning.

Our experimental approach was framed within the perspective of dual reinforcement learning (RL) systems. Here, an extensive body of psychological and neuroscientific literature indicates that behaviour is governed by two distinct systems[8–22]—a rigid, retrospective model-free (MF) system[23,24] and a flexible, prospective model-based (MB) system[23,25]. Unlike the MF system, which tends to repeat actions that were successful in the past, the MB system deliberates upon the likely future effects of potential actions. Recent findings suggest that when making decisions, an agent's behaviour reflects contributions from both systems[16,25]. A range of theories highlights diverse principles underlying dual system interactions, such as speed accuracy trade-offs[26], an actor-trainer dichotomy[17,27], reliability-based arbitration[9,28] and a plan-to-habit strategy[29]. A separate rich body of research shows that when RL occurs in the face of state uncertainty, beliefs about states underlie the calculation of prediction errors and guide learning[1–7].

Here we develop and test a theory of retrospective MB inference. We propose that, in addition to prospective planning, a MB system performs retrospective inference about states in order to resolve uncertainty operative at the actual time of a decision. We can summarise this as posing a question that asks not only "where should I go?" but also "where did I come from?" Our theory draws inspiration from a dual role that cognitive models play in real-life not only in predicting future states and outcomes, but also in retrospectively inferring past hidden states. In a social domain, for example, one relies on models of close others not only to predict their future actions but also to cast light on motives that underlined their past actions. Our key proposal is that in situations involving temporary state uncertainty, a MB system exploits its model of task structure to resolve uncertainty retrospectively, i.e., following a choice. Furthermore, we suggest that a MF system can exploit the outcome of a MB inference to assign the credit from a choice preferentially to an inferred state, thus underpinning a form of interaction between the MB and MF systems.

To test this hypothesis we designed a task with state uncertainty and in which it was possible to dissociate MB and MF control of behaviour. At selected points, subjects had to choose between pairs of bandits (i.e., lotteries), and were informed that only one of the two bandits of their chosen pair will be executed. Critically, observers were not informed explicitly which of the two bandits this actually was but they could retrospectively infer the identity of the executed bandit once they observed which outcomes ensued. We found evidence for MF learning that was greater for the executed bandit, supporting a hypothesis that a MB system retrospectively infers the correct state and that this inference directs a MF credit assignment. In our discussion we consider potential algorithmic and mechanistic accounts of these findings.

## Results

**Behavioural task.** We developed a dual-outcome bandit task in which we introduced occasional instances of uncertainty with respect to which bandit was actually executed. In brief, at the outset, participants were introduced to a treasure castle that contained pictures of four different objects. Subjects were trained as to which pair of rooms—out of four castle rooms characterised by different colours—each object would open. Each individual room was opened by two distinct objects, while each object opened a unique pair of rooms (Fig. 1a).

Following training, each participant played 504 bandit trials. Two out of every three trials were "standard trials", in which a random pair of objects that shared a room as an outcome were offered for 2 s. Subjects had to choose between this pair. Once a choice was made, the two corresponding rooms opened, one after the other, and each could be either empty or contain a bonus point worth of treasure (Fig. 1b). Every third trial was an "uncertainty trial" in which, rather than two objects, two disjoint pairs of objects were presented for choice. Crucially, each of these presented pair of objects shared one Common-room outcome. Participants were informed that a "transparent ghost" who haunted the castle would randomly nominate, with equal probability, one of the objects within this chosen pair. Since the ghost was transparent, participants could not see which object it nominated. Nevertheless, in this arrangement subjects still visited the opened rooms and collected any treasure they found. Importantly, when the ghost nominated an object, the room Common to both objects in the chosen pair opened first, and the room that was Unique to the ghost-nominated object opened second. Thus, it was at this second point that participants could infer the object that the ghost had nominated. Across the time-course of the experiment, room reward probabilities were governed by four independently drifting random walks (Fig. 1c).

**Model-free and model-based contributions in standard trials.** Because our main analysis concerns the effect of MB inference on MF-learning, it is important to show first that participants in the task relied on contributions from both systems. Thus, we begin by specifying how each putative system contributes to performance and, in so doing, show significant MF and MB contributions to choices.

The MF system caches the rewards associated with previous object choices. In the choice phase of standard trials, the MF system feeds retrieved $Q^{MF}$-values of the objects offered for choice into a decision module. In the learning phase on standard trials, the simplest form of MF system performs a Rescorla–Wagner[30] update with learning-rate lr, based on a prediction error corresponding to the total reward of the two consequent rooms (ignoring the sequence):

$$Q^{MF}(\text{chosen object}) \leftarrow Q^{MF}(\text{chosen object}) + \text{lr} * (\text{total reward} - Q^{MF}(\text{chosen object}))$$

(1)

Importantly, as the MF system lacks a model of the task transitions, on each trial its credit-assignment is restricted to updates generated by the object chosen on that trial (as in Doll et al., 2015[25]). For example, a blue-room reward following a choice of the stove will update the MF $Q^{MF}$-value of the stove but not of the light-bulb, which also happens to open the blue room.

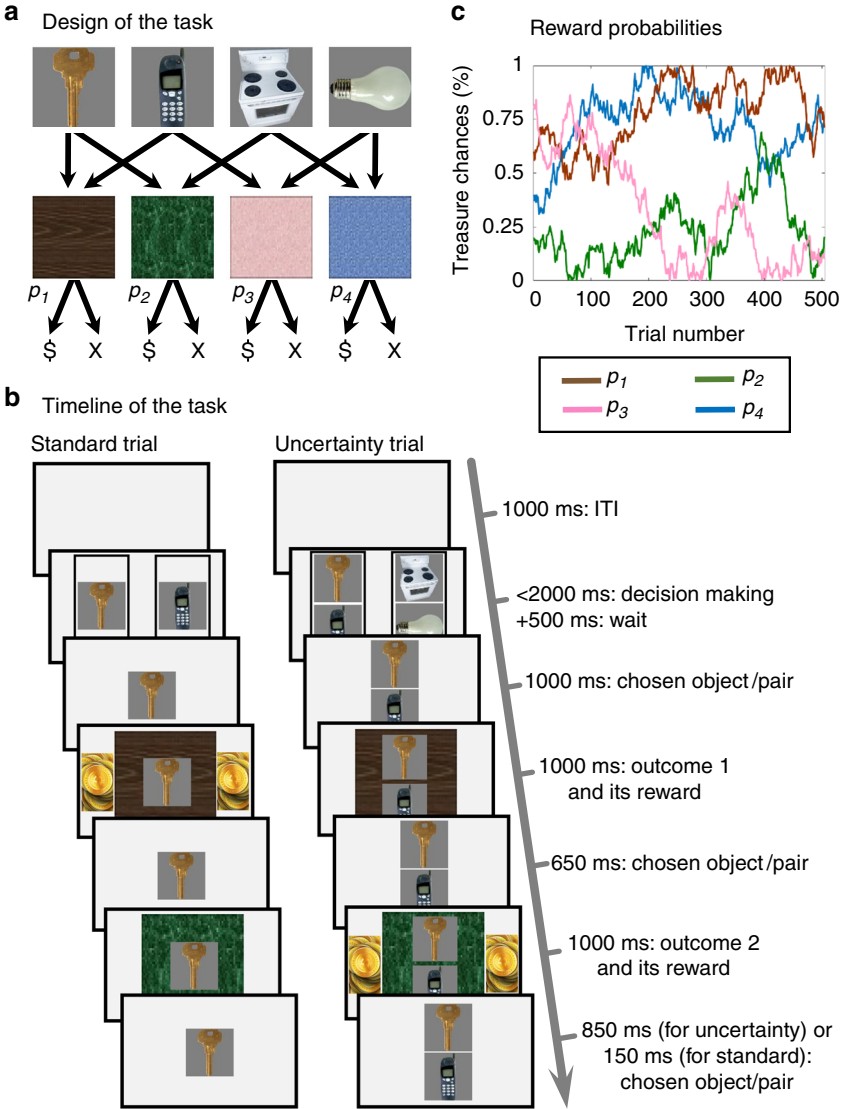

**Fig. 1** Task Structure. **a** Participants were introduced to four objects and learned which unique pair of rooms each object could open. Each room was opened by either of two different objects. When opened, rooms probabilistically provided a one-point value treasure. **b** During standard trials (left) participants were asked to choose, within 2 s, one of two randomly-offered objects, and then visited sequentially the pair of rooms opened by that chosen object. At this point, participants discovered, for each room, whether it contained a treasure or was empty. In every third, uncertainty, trial (right), participants chose one of two object-pairs (left or right). Participants were instructed that a hidden hand (a ghost) would nominate one of the objects in their chosen pair randomly without revealing which. Participants then visited the pair of rooms that the object, nominated by the ghost, opened and earned whatever treasure was available in those rooms. Importantly, on an uncertainty trial, the first outcome was common to both objects in the chosen pair and hence did not support an inference about the ghost's nomination. The second outcome, however, was unique to the ghost-nominated outcome and hence, allowed an MB retrospective inference with respect to the ghost's nomination. We hypothesized that participants not only inferred the ghost's nomination but that in addition this inference would guide the expression of MF learning. This in turn leads to a specific prediction that MF learning will be stronger for a ghost-nominated object compared to a ghost-rejected object. **c** Across trials, reward probabilities for the four rooms drifted according to independent Gaussian random walks with reflecting bounds at 0 and 1. Images adapted from the stimulus set of Kiani et al. 2007, ref. [40]

There are various possible MB systems. The most important difference between them concerns whether the MB system learns directly about the rooms, and uses its knowledge of the transition structure to perform an indirect prospective calculation of the values of the objects presented on a trial based on the values of the rooms to which they lead (henceforth, a 'room value learning' MB system), or whether it uses knowledge of the transition structure to learn indirectly about the objects, and then uses these for direct evaluation (henceforth, an 'object value learning' MB system). While these two formulations of an MB system are similar in that they both allow generalization of observations about rewards to objects that were not chosen or nominated, they nevertheless differ in their value calculations and generate different experimental predictions.

Until the very end of this section, our presentation relies on the room value learning formulation. This is justified because a model comparison (Supplementary Fig. 5) revealed that it was superior to an object-value learning formulation. According to this model, rather than maintaining and updating $Q^{MB}$-values for the objects, an MB system instead does so for the rooms, and prospectively calculates on-demand $Q^{MB}$-values for the offered objects. In standard trials, this is (normatively) based on the

arithmetic sum of the values of their corresponding rooms:

$$Q^{MB}(object) = Q^{MB}(corresponding\ room\ 1) + Q^{MB}(corresponding\ room\ 2) \quad (2)$$

During the learning phase of standard trials, the system performs Rescorla–Wagner updates for the values of the observed rooms:

$$Q^{MB}(room) \leftarrow Q^{MB}(room) + lr * (room\ reward - Q^{MB}(room)) \quad (3)$$

Consequently, unlike MF, MB credit-assignment generalizes across objects that share a common outcome. To continue the example, when a blue room is rewarded, $Q^{MB}$(blue room) increases and in following calculations, the on-demand $Q^{MB}$-values for both the stove and light-bulb will benefit.

We next show, focusing on model-agnostic qualitative behavioural patterns alone, both MF and MB contributions to choices. These analyses are accompanied by model-simulations of pure MF and pure MB contributions to choices. The main purpose of these simulations is to support the reasoning underlying our analyses. A full description of these models is deferred to a later section.

In this behavioral analysis we confine ourselves to Standard→ Standard trial transitions. Consider a trial-$n + 1$, which offers for choice the trial-$n$ chosen object (e.g. key), against another object (e.g. stove; Fig. 2a). The two offered objects open a "Common room" (e.g. green, denoted C) but the trial-$n$ chosen object also opens a "Unique room" (brown, denoted U). We tested if the probability of repeating a choice depended on the trial-$n$ Common-room outcome, while controlling for the Unique-room outcome (Fig. 2b). From the perspective of the MB system (Fig. 2c), the value of the Common room, and in particular, whether it had been rewarded on trial-$n$, exerts no influence on the relative $Q^{MB}$-values of the currently offered objects, because this value "cancels out". For example, the calculated MB value of the key on trial-$n + 1$ is the sum of the MB Q values of the green and brown rooms. Similarly, the calculated MB value of the stove on trial-$n + 1$ is the sum of the MB Q values for the green and the blue rooms. The MB contribution to choice depends only on the contrast between these calculated key and stove values, which equals the difference between the MB values of the brown and blue rooms. Notably, the value of the green room is absent from this contrast and hence does not affect MB contributions to choice on trial-$n + 1$. From the perspective of the MF system, however, a Common-room reward on trial-$n$ reinforces the chosen object alone leading to an increase in its repetition probability, as compared to non-reward for this Common room (Fig. 2d).

Using a logistic mixed effects model, in which we regressed the probability of repeating the trial-$n$ choice on Common and Unique trial-$n$ outcomes, we found (Fig. 2b) a main effect for the Common outcome ($b = 0.98$, $t(3331) = 8.22$, $p = 2.9e{-}16$; this is evident in the red, C-Rew, line being above the blue, C-Non, line), supporting an MF contribution. Additionally, we found a main effect for the Unique reward ($b = 2.03$, $t(3331) = 14.8$, $p = 4.9e{-}48$; evident in the increase of both red and blue lines from U-Non to U-Rew), as predicted by both MF and MB contributions, and a significant interaction between Common and Unique outcomes ($b = 0.46$, $t(3331) = 2.49$, $p = 0.013$) indicating that the effect of the Common outcome was modestly larger when the Unique room was rewarded than unrewarded. An analysis of simple effects revealed that the Common room had a positive effect irrespective of whether the Unique room was unrewarded ($b = 0.75$, $t(3331) = 4.60$, $p = 4e{-}6$) or rewarded ($b = 1.21$, $t(3331) = 8.73$, $p = 4e{-}18$; See Supplementary Note 1 for clarifications about Fig. 2).

Turning next to a MB contribution, consider a trial-$n + 1$, which excludes the trial-$n$ chosen object (e.g., key; Fig. 2f) from the choice set. In this case, the trial-$n$ chosen object shares a Common room (e.g. green) with only one of the trial-$n + 1$ offered objects (e.g. stove), whose choice we label a "generalization". Additionally, the trial-$n$-chosen object shares no outcome with the other trial-$n + 1$ offered object (e.g. bulb). We examined whether the probability of generalizing the choice depended on the Common outcome on trial-$n$ (Fig. 2g). A MB contribution (Fig. 2h) predicts a higher choice generalization probability when the Common-room was rewarded on trial-$n$, as compared to non-rewarded, because this reward increases the calculated Q-values of all objects (including the stove) that open that room.

Considering the MF system (Fig. 2i), trial-$n$ reward-events cannot causally affect choices on trial-$n + 1$, because learning on trial-$n$ was restricted to the chosen object, which is not present on trial-$n + 1$. However, MF predictions are somewhat complicated by the fact that a Common green outcome on trial-$n$ (reward vs. non-reward) is positively correlated with the MF Q-value of the stove on trial-$n + 1$. To understand this correlation, note that the reward probability time series for each room is auto-correlated since it follows a random walk. This means coarsely that the green room's reward probability time series alternates between temporally extended epochs during which the green room is better or worse than its average in terms of a reward probability. When the green room is rewarded vs. unrewarded on trial-$n$, it is more likely that the green room is currently spanning one of its better epochs. Importantly, this also means that the stove was more likely to earn a reward from the green room when it had recently been chosen prior to trial-$n$. Thus, a Common-room reward for the key on trial-$n$ i$s$ positively correlated with the MF value of the stove on trial-$n + 1$. It follows that an MF contribution predicts a higher generalization probability when the Common room is rewarded as compared to non-rewarded (Fig. 2i). Critically, because this MF prediction is mediated by the reward probability of the Common room (i.e., how good the green room is in the current epoch), a control for this probability washed away a MF contribution to the effect of the Common trial-$n$ outcome on choice generalization (Fig. 2m), hence implicating the contribution of an MB system (Fig. 2l).

A logistic mixed effects model showed (Fig. 2k) a positive main effect for the Common outcome ($b = 0.40$, $t(3225) = 3.328$, $p = 9e{-}4$) on choice generalization, supporting an MB contribution to bandit choices. Additionally, we found a significant main effect for the Common outcome's reward probability ($b = 1.94$, $t(3225) = 7.08$, $p = 2e{-}12$) as predicted by both systems (Fig. 2l, m) and no interaction between the Common trial-$n$ outcome and the Common room's reward probability ($b = -0.75$, $t(3225) = -1.76$, $p = 0.079$). Note that unlike our analysis which pertained to an MF contribution (Fig. 2a–e), the analysis pertaining to MB contributions did not control for the effect of the Unique room (e.g. brown), because it was an outcome of neither choice object on trial-$n + 1$. Hence, this room's outcome was expected to exert no influence on choice generalization from the perspective of either MB or MF. Indeed, when we added the Unique-room trial-$n$ outcome to the mixed effect model, none of the effects involving the Unique-room outcome were significant (all $p > 0.05$), and the effects of the Common room's outcome and Common reward probability remained significantly positive (both $p < 0.001$) with no interaction. Consequently, this room was not considered in our analyses pertaining to a MB's contribution to performance.

**MB inference guides MF learning on uncertainty trials.** Having established that both systems contribute to performance we next addressed our main set of questions: do people infer the ghost-

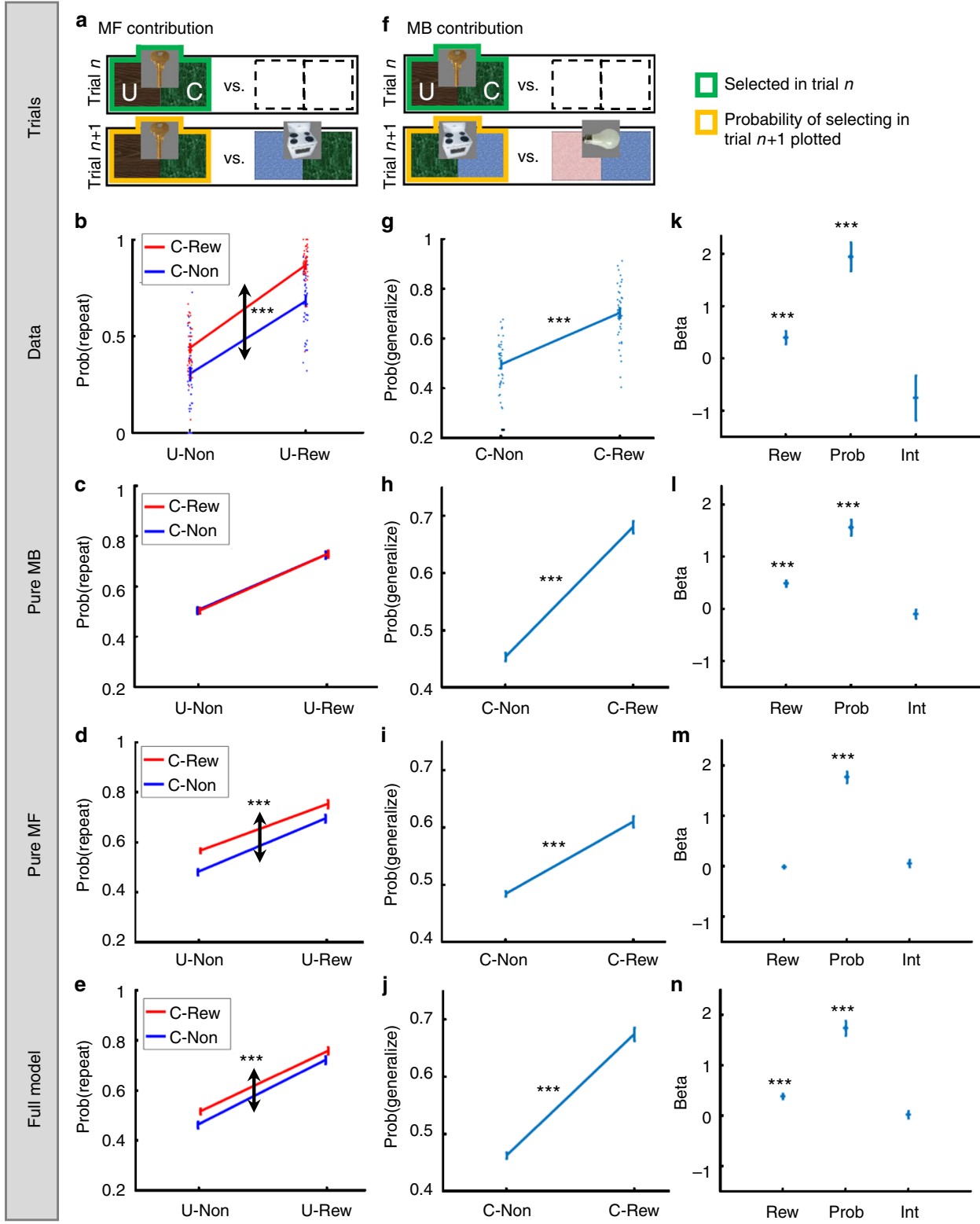

nominated object and, if so, does this inference guide the expression of MF learning? To address these questions we probed whether on uncertainty trials, MF learning discriminated between constituent chosen-pair objects, a finding that would implicate a retrospective inference of the ghost-nominated object. MF learning in this instance was gauged by probing on uncertainty trials the effect of outcomes on follow-up standard trial choices (Fig. 3).

Recall that following a ghost nomination, agents observe first the room that is common to the objects in their chosen pair, and then the room unique to the ghost-nominated object. Therefore, agents have a 50–50% belief with respect to the ghost-nominated object when they choose a pair, and this belief is maintained upon observing the first outcome, which is non-informative with respect to inference of the ghost's nominee. Critically, following the second outcome, an MB system can infer the ghost-

**Fig. 2** MF and MB contributions to performance. **a** In showing a contribution of the MF system, we analysed only standard trials that followed a standard trial and that offered for choice the previously chosen object. For clarity, we represent objects by their associated pair of rooms; in the experiment participants saw the object images alone. Places for objects whose identity did not affect the analysis (and were marginalized over) are left empty. **b** The empirical probability of repeating a choice as a function of the previous-trial Common ("C"), and Unique ("U") outcomes. The main effect of the Common outcome highlights an MF contribution to bandit choices. **c** Indeed, a pure MB model failed to predict this effect (our models are described in the computational modelling section), **d** but a pure MF-action model predicted it. **e** The full model predicted this effect. **f** In showing a contribution of the MB system, we analysed only standard trials that followed a standard trial and that excluded the previously chosen object. **g** The empirical main effect of the Common room on generalization probability. **h–j** The pure MB, pure MF and the full model, all predicted a positive effect for the Common room. **k** The empirical coefficients of the trial-$n$ Common room's outcome (unrewarded vs. rewarded; Rew), reward probability (Prob) and their interaction (Int) when choice generalization is regressed on these variables. The positive coefficient of the Common outcome highlights a MB contribution to bandit choices. **l** Indeed, a pure MB model predicted a positive coefficient for the Common reward, **m** whereas a pure MF-action model did not. **n** The full model predicted a positive coefficient for the Common outcome. Error bars correspond to SEM across-participants calculated separately in each condition ($n = 40$). Dotted arrows indicate the main effect of focal interest. *,** and *** denote $p < 0.05$, $p < 0.01$ and $p < 0.001$, respectively. When no asterisk appears, the effect of interest is non-significant ($p > 0.05$). In **g–j** $p$-values were calculated based on paired-samples $t$-tests. In panels **b–e**, **k–n** $p$-values were calculated based on mixed effects logistic regression models. Dots in panels **b**, **g** correspond to individual participant results. Images adapted from the stimulus set of Kiani et al. 2007, ref. [40]

nominated object, with perfect certainty, based upon a representation of task transition structure. The second outcome is therefore informative with respect to inference of the ghost's nominee. Henceforth, we denote the first and second outcomes by "N" (for Non-informative) and "I" (for Informative). We hypothesised that inferred object information is shared with the MF system and this directs learning towards the chosen object. Here we assume that after outcomes are observed, the MF system updates the $Q^{MF}$-values of both objects in the chosen pair, possibly to a different extent (i.e., with different learning rates). In the absence of inference about the ghost's nomination, the MF system should update the $Q^{MF}$-values of both objects to an equal extent. Thus, a finding that learning occurred at a higher rate for the ghost-nominated, as compared to the ghost-rejected, object would support a retrospective inference hypothesis.

Consequently, we examined MF learning in uncertain trials by focusing on Uncertain→Standard trial transitions. The task included three sub-transition types, which were analysed in turn. We first present the findings in support of our hypothesis. In the discussion we address possible mechanistic accounts for these findings.

**Informative outcome credit assignment to nominated object**. First, we show that MF assigns credit from the Informative outcome to the ghost-nominated object. Consider a standard trial-$n$ + 1 (following an uncertainty trial-$n$) that offered for choice the ghost-nominated object (e.g. key) alongside an object (e.g. phone) from the trial-$n$ non-selected pair that shared the previously inference-allowing, I, outcome (e.g. brown) with the ghost-nominated object; (Fig. 4a). We label such trials "repeat trials". A choice repetition is defined as a choice of the previously ghost-nominated object. We tested whether a tendency to repeat a choice depended on the trial-$n$ Informative outcome. Note that from the perspective of MB evaluations on trial-$n$ + 1 the Informative room's value cancels out because this room is associated with both offered objects. From the perspective of the MF system, however, if Informative outcome credit was assigned to the ghost-nominated object on trial-$n$, then reward vs. non-reward on an Informative room should increase the repetition probability.

A logistic mixed effects analysis, in which we regressed the probability of repeating the choice on trial-$n$ outcomes, showed a main effect for the Informative (I) outcome ($b = 0.84$, $t(2204) = 7.17$, $p = 1e–12$), supporting an hypothesis that MF assigns credit from this outcome to the ghost-nominated object. Additionally, we found a main effect for the Non-informative (N) outcome ($b$

= 1.49, $t(2204) = 10.63$, $p = 9e–26$), as predicted by both MF credit-assignment to the ghost-nominated object and by MB contributions, and no significant interaction between the Informative and Non-informative outcomes ($b = 0.35$, $t(2204) = 1.66$, $p = 0.098$).

**Informative outcome credit assignment to rejected object**. Second, we tested whether credit from the Informative outcome was also assigned by an MF system to the ghost-rejected object. Consider a standard trial-$n$ + 1 that offered for choice the ghost-rejected object (e.g. stove) alongside an object from the trial-$n$ non-selected pair (e.g. bulb) that shares an outcome with the ghost-rejected object (Fig. 4b). We label such trials "switch trials". A choice generalization is defined as a choice of the previously ghost-rejected object. Does the tendency to generalize a choice depend on the trial-$n$ Informative outcome? An MB contribution predicts no effect, because the Informative room is an outcome of neither trial-$n$ + 1 choice-object. From the perspective of the MF system, however, if credit had been assigned to the ghost-rejected object on trial-$n$, then reward versus non-reward on an Informative outcome should increase the generalization probability.

A logistic mixed effects model, in which we regressed the choice generalization probability on the trial-$n$ outcome, showed a main effect for the Informative outcome ($b = 0.32$, $t(2207) = 2.92$, $p = 0.004$), supporting the hypothesis that an MF system assigns credit to the ghost-rejected object. This result is striking because it shows that the MF system assigned credit from the Informative outcome to an object that is not related to that outcome. This challenges any notion of perfect MB guidance of MF credit-assignment. However, it is consistent with the possibility that some participants, at least some of the time, do not rely on MB inference because when MB inference does not occur, or when it fails to guide MF credit-assignment, the MF system has no basis to assign credit unequally to both objects in the selected pair. Additionally, we found a main effect for the Non-informative outcome ($b = 1.15$, $t(2207) = 8.80$, $p = 3e–18$), as predicted, not only by an MF credit-assignment to the ghost-rejected object account but also by MB contributions. We found no significant interaction between the Informative and Non-informative outcomes ($b = -0.07$, $t(2207) = -0.31$, $p = 0.755$).

**Informative outcome preferential credit assignment**. Hitherto we showed that on uncertainty trials, credit obtained from an Informative, inference-allowing, outcome was assigned in a MF manner to both the ghost-nominated and the ghost-rejected

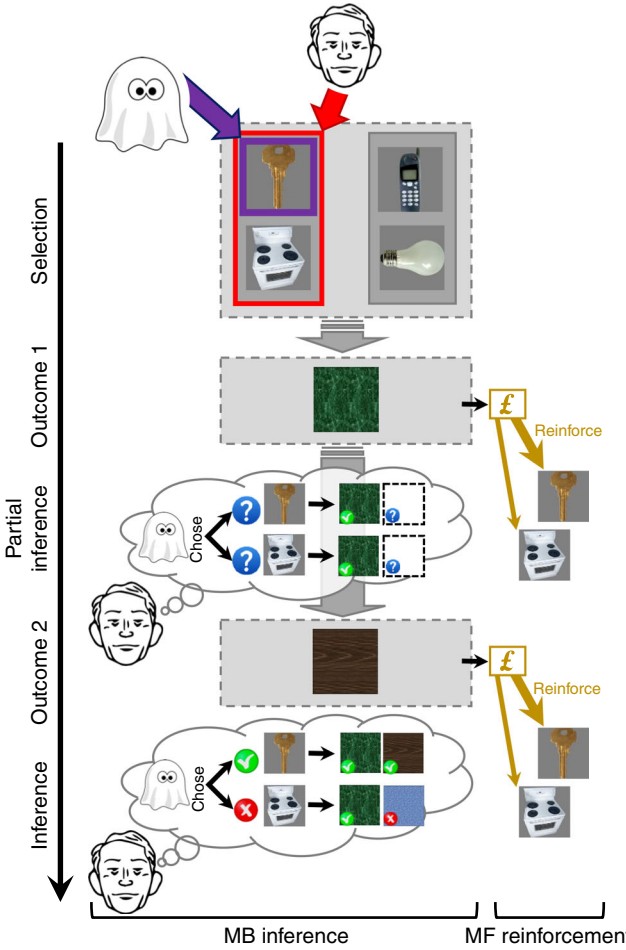

**Fig. 3** MB retrospective inference and its effects on MF learning. After the agent choses an object pair (here, left), the ghost nominates randomly (and latently) one of the objects in this selected pair (here, top). The agent, therefore, has a 50–50% belief with respect to the ghost-nominated object. The posterior belief after observing the first (green) outcome remains 50–50% since this outcome is common to both chosen-pair objects. The second (brown) outcome is unique to the ghost-nominated object and hence, a model of the transition structure allows for perfect inference on the ghost's nomination. We probed different types of standard follow-up trials to examine how the MF system assigned credit to rewards that were obtained on the uncertainty trials. In the absence of any inference about the ghost's nomination, learning should be equal in strength for both objects in the pair chosen by the subject. Supporting our retrospective-inference theory, we found that an MF assignment of credit received from the second outcome is mainly attributed to the ghost-nominated object (indicated by the thickness of the bottom "reinforcement" arrows). Strikingly, we also found that credit from the first outcome was assigned to a larger extent to the ghost-nominated object (indicated by the thickness of the top "reinforce" arrows). Images adapted from the stimulus set of Kiani et al. 2007, ref. [40]

objects. We hypothesized that the Informative outcome would support MB retrospective inference, and boost MF learning for the ghost-nominated object. Thus, we compared (Fig. 4d) effects that the Informative outcome exerted over choice on follow-up repeat and switch trials, i.e. the effects from the previous two analyses (Fig. 4a, b). For each participant, we calculated the contrast for the probability of repeating a choice in repeat follow-ups when the Informative trial-$n$ outcome was rewarded vs. non-rewarded ($M = 0.159$, SE $= 0.024$).

Additionally, we computed the corresponding contrast for the probability of generalizing the choice in switch follow-ups ($M = 0.086$, SE $= 0.024$). Importantly, the former contrast was larger than the latter ($t(39) = 2.29$, $p = 0.028$), implicating enhanced MF credit-assignment to the ghost-nominated object.

**Non-informative outcome preferential credit assignment**. We next examine MF credit assignment for the first Non-informative outcome. Consider a standard trial-$n + 1$ that offered for choice the trial-$n$ ghost-nominated object alongside the ghost-rejected object, i.e. the trial-$n$ chosen pair (Fig. 4c). We label such trials "clash trials". We defined a choice repetition as a trial-$n + 1$ choice of the object the ghost-nominated on trial-$n$. As in previous cases, the MB contribution predicts no effect of the trial-$n$ Non-informative outcome due to cancelling out. From the perspective of the MF system, however, if credit was assigned preferentially to the ghost-nominated object on trial-$n$, then a Non-informative-reward, as compared to non-reward, should increase the repetition probability.

A logistic mixed effects model, in which we regressed the choice repetition probability on trial-$n$ outcomes, showed a main effect for the Non-informative outcome ($b = 0.20$, $t(2197) = 1.98$, $p = 0.048$), supporting the hypothesis that MF credit assignment is mainly directed to the ghost-nominated object. This finding is striking, because during reward administration for the first room, participants have a 50–50% belief about which object had been nominated by the ghost. We suggest that this supports the idea of an MF credit-assignment mediated by a later retrospective-inference. Additionally, we found a main effect for the Informative outcome ($b = 1.40$, $t(2197) = 9.41$, $p = 1e-20$), as predicted by both the enhanced MF credit-assignment for the ghost-nominated object hypothesis and by a MB contribution. We found no significant interaction between Non-informative and Informative outcomes ($b = 0.48$, $t(2197) = 1.89$, $p = 0.059$) (See Supplementary Fig. 1 for supporting model simulations demonstrating that preferential credit-assignment for the ghost-rejected object is necessary to account for the empirical effects).

**Computational modelling**. One limitation of the analyses reported above is that they isolate the effects of the immediately preceding trial on a current choice. However, the actions of RL agents are influenced by the entire task history. To account for such extended effects on behavior, we formulated a computational model that specified the likelihood of choices. The model allowed for a mixture of contributions from MB and MF processes. Critically, our model included three free MF learning-rate parameters, which quantified the extent of MF learning for standard trials ($lr_{standard}$), for the ghost-nominated object in uncertainty trials ($lr_{ghost-nom}$) and for the ghost-rejected object on uncertainty trials ($lr_{ghost-rej}$). Additionally, we formulated four sub-models of interest: (1) a pure MB model, which was obtained by setting the contribution of the MF and its learning rates to 0 (i.e. $w_{MB} = 1$; $lr_{standard} = lr_{ghost-nom} = lr_{ghost-rej} = 0$), (2) a pure MF-action model, which was obtained by setting the contribution of the MB system to choices and its learning rate to 0 (i.e. $w_{MB} = 0$; $lr_{MB} = 0$; Note that in this model, a MB inference was allowed to guide MF inference), (3) a 'non-inference' sub-model obtained by constraining equality between the learning rates for the ghost-nominated and rejected objects, $lr_{ghost-nom} = lr_{ghost-rej}$ and (4) a 'no-learning for ghost-rejected' sub-model, which constrained the learning rate for the ghost-rejected object to $lr_{ghost-rej} = 0$. We fitted these models to each participant's data using a Maximum-Likelihood method (See the methods for full details about the models; See Supplementary Table 1 for the full model's fitted parameters).

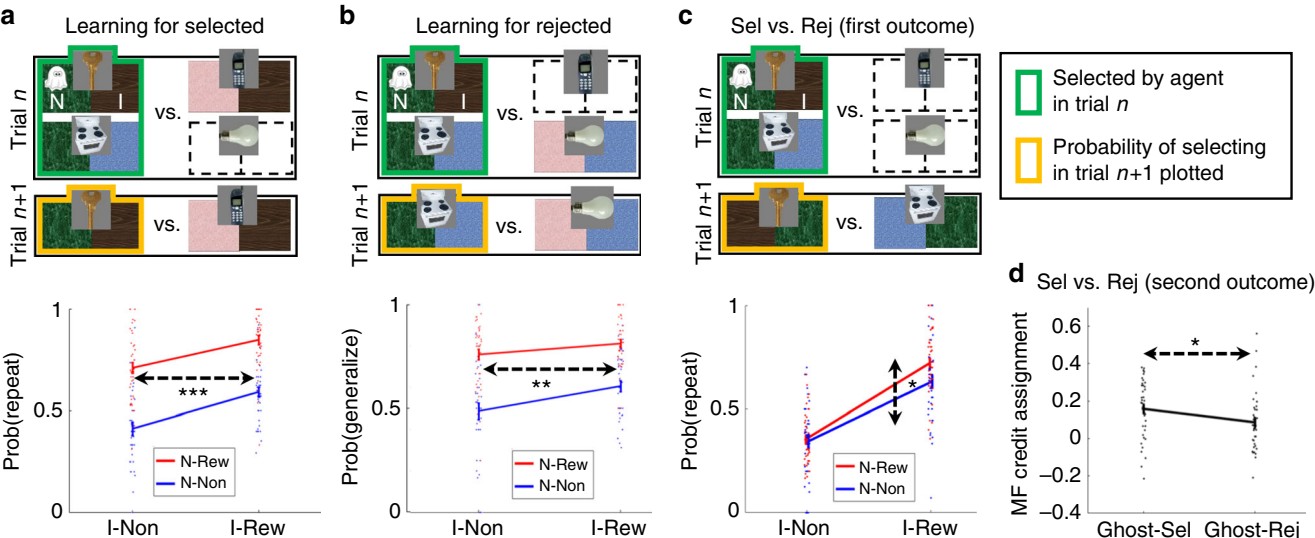

**Fig. 4** MF Learning for ghost-nominated and ghost-rejected objects on uncertainty trials. **a** The probability of repeating a choice, i.e., select the trial-*n* ghost-nominated object, as a function of the previous-trial non-informative, "N" (here green), and informative, inference-allowing, "I" (here brown) outcomes (bottom). Only "repetition" standard trials that offered for choice the previously ghost-nominated object alongside the object from the previously non-chosen pair, which shared the previously informative outcome with the ghost-nominated object, were analysed. The main effect of the Informative outcome implies that credit from this outcome was assigned by a MF learner to the ghost-nominated object. **b** Similar to **a**, but here, the probability to generalize the choice, i.e., select the ghost-rejected object is shown. Only "switch" standard trials that offered for choice the previously ghost-rejected object alongside an object from the previously unchosen pair (the one that shares an outcome with the ghost-rejected object) were analysed. The main effect of the Informative (brown) outcome implies that credit from this outcome was assigned by MF to the ghost-rejected object. **c** Similar to **a** but only "clash" standard trials that offered for choice the previously ghost-nominated and rejected objects, i.e., the previously chosen pair, were analysed. The main effect of the non-Informative outcome (green) implies that credit from this first outcome was assigned by MF mainly to the ghost-nominated object. Each standard trial after an uncertain trial was either a repeat, switch or a clash trial and hence contributed to exactly one of the panels. **d** Comparing the main effects from the analyses in **a** and **b** shows that credit from the second informative, inference-allowing, outcome was assigned by MF mainly to the ghost-nominated object. Error bars correspond to SEM across-participants calculated separately in each condition, $n = 40$. Dotted arrows indicate the main effect of interest. *,** and *** denote $p < 0.05$, $p < 0.01$ and $p < 0.001$, respectively. In case no asterisks are presented, the effect of interest was not significant. In **a–c** *p*-values were calculated based on a mixed effects logistic regression models. In **d**, *p*-values were calculated based on a paired-sample *t*-test. Dots represent individual participant results. Images adapted from the stimulus set of Kiani et al. 2007, ref. [40]

We next compared our full model separately with each of these sub-model. Our model comparisons were all based on a bootstrap generalized-likelihood ratio test (BGLRT[31]) between the full model and each of its sub-models in turn (see methods for details). In brief, this method is based on hypothesis testing, where, in each of our model comparisons, a sub-model serves as the H0 null hypothesis and the full model as the alternative H1 hypothesis. The results were as follows. First, we rejected the pure MB and the pure MF-action sub-models for 26 and 34 individuals, respectively, at $p < 0.05$, and at the group level (both $p < 0.001$) (Fig. 5a, b). These results support a conclusion that both MB and MF systems contribute directly to choices in our task. Next, we rejected the 'no-learning for ghost-rejected' sub-model for 10 individuals at $p < 0.05$, and at the group level ($p < 0.001$) (Fig. 5c), showing that in uncertainty trials, learning occurs for ghost-rejected objects. Additionally, and most importantly, we rejected the 'non-inference' sub-model for 12 individuals at $p < 0.05$, and at the group level ($p < 0.001$) (Fig. 5d), showing that learning is different for the retrospectively inferred ghost-nominated than the ghost-rejected object. We note that although the 'non-inference' and the 'no learning for ghost rejected' models were each rejected for a minority of the participants (10 and 12 participants, respectively), the size of these minorities are nevertheless substantial considering that our task was not optimally powered to detect individual participant effects, and given significance testing at $p = 0.05$ should yield on average two rejections (out of 40 participants) when the null hypothesis holds for all participants. Finally, applying the

Benjamini–Hochberg procedure[32] to control for the false discovery rate, the 'non-inference' and the 'no learning for ghost rejected' models were rejected for 10 and four individuals, respectively.

We next ran a mixed effects model in which we regressed the MF learning rates from the full model on the learning context (standard/ghost-nominated/ghost-rejected). This analysis (Fig. 6a) showed that the three learning rates differed from each other ($F(2,117) = 3.43$, $p = 0.036$). Critically, as expected the learning rate for the ghost-nominated object was greater than for the ghost-rejected object ($F(1,117) = 6.83$, $p = 0.010$). Additionally, the learning rate for the standard condition was larger than for the ghost-rejected object ($F(1,117) = 4.05$, $p = 0.047$), with no significant difference between the learning rate for the ghost-nominated object and for standard trials ($F(1,117) = 0.920$, $p = 0.340$). These findings provide additional support for the hypothesis that a retrospective inference process directs MF learning towards the object that could be inferred to have been nominated, and away from the one that was rejected.

Finally, because the models reported in the main text did not include MF eligibility traces[33], we examined whether such traces, rather than preferential learning based on retrospective inference, might account for the qualitative "preferential MF credit assignment" patterns presented in Fig. 4c, d. We found that models based on eligibility-driven MF learning failed to account for the observed patterns (See methods for full model descriptions; See Supplementary Fig. 1I–L for the predictions of these models).

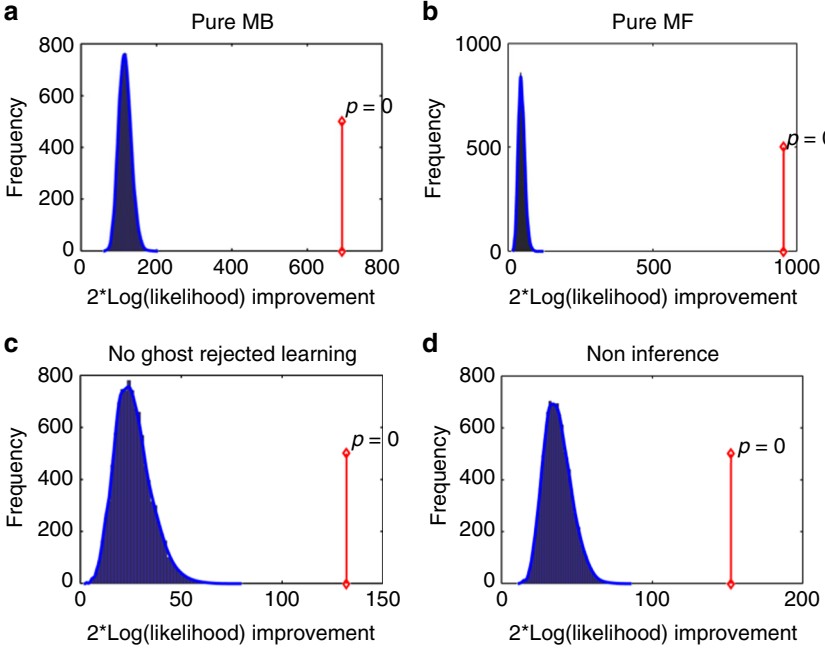

**Fig. 5** Model-comparison results. **a** Results of the bootstrap-GLRT model-comparison for the pure MB sub-model. The blue bars show the histogram of the group twice log-likelihood improvement (model vs. sub-model) for synthetic data that was simulated using the sub-model (10001 simulations). The blue line displays the smoothed null distribution (using Matlab's "ksdensity"). The red line shows the empirical group twice log-likelihood improvement. Zero out of the 10001 simulations yielded an improvement in likelihood that was at least as large as the empirical improvement. Thus, the sub-model can be rejected with $p < 0.001$. **b** Same as **a**, but for the pure MF-action sub-model. This sub-model was rejected with $p < 0.001$. **c** Same as **a** but for the no learning for ghost-rejected sub-model, $p < 0.001$. **d** Same as **a** but for the non-inference sub-model, $p < 0.001$

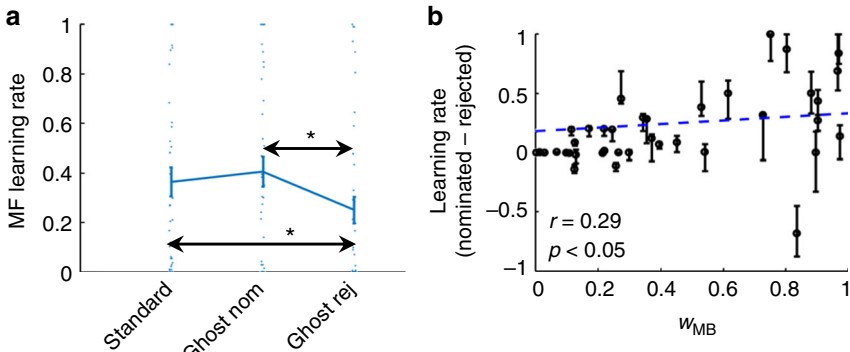

**Fig. 6** Analyses based on the ML parameters of the full model. **a** Group averaged MF learning rates for standard trials, ghost-nominated and the ghost-rejected object. Error bars represent standard errors, $n = 40$. * denotes $p < 0.05$ according to a mixed effects linear model. Dots represents individual participants. **b** The specificity of MF learning, i.e. the contrast between learning rates for the ghost-nominated and rejected objects (ordinate), is plotted against the relative MB contribution (abscissa). Each dot corresponds to a single participant. The dashed line is the regression line. Error bars represent the densest interval that contained 50% of the mass of the estimated learning-rate difference distribution obtained by parametrically bootstrapping the data (see methods for details). The $p$-value was calculated based on a one-sided permutation test

**Correlation between MB contribution and learning specificity.** As model-basedness increases, the relative contribution of MF to performance, and hence the influence of MF's learning rates on performance, decreases. Importantly, the full model allowed for an estimation of the extent to which each subject's MF credit assignment prefers the ghost-nominated over the ghost-rejected object ($lr_{ghost-nom} - lr_{ghost-rej}$), controlling for the extent to which that subject relies on MB in his/her choices ($w_{MB}$). Our retrospective-inference theory predicts that MB inference of the ghost-nominated object, which relies on knowledge of the task's transition structure, directs MF learning towards this inferred object. This suggests that the greater the degree of a subject's model-basedness, the better they can infer the ghost's

nominations and hence, the more specific MF-learning might be on uncertainty trials. In support of this hypothesis, we found (Fig. 6b) a positive across-participants correlation ($r = 0.29$, $p = 0.034$ based on one-sided permutation test) between $w_{MB}$ and $lr_{ghost-nom} - lr_{ghost-rej}$, as quantified by the maximum-likelihood model-parameters (Supplementary Fig. 7 for a further control analysis, addressing the higher learning-rate measurement noise as model-basedness increases).

**Object value learning MB system.** As noted, all the main analyses in this section are based on a room value learning version of MB reasoning. This was motivated by its superior fit to the data.

However, in order to support the robustness of our results we repeated our analyses of the influence of MB inference on MF value updating, but now assuming that an object-value learning version was responsible for the MB values (noting that the MB inference about which object had been nominated by the ghost is unaffected). In this approach we obtained the same conclusions (Supplementary Figs. 2–4 and 6).

## Discussion

Extensive research in dual systems RL has emphasized a fundamental "temporal-orientation" distinction between the MF and MB system. According to this notion, a MF system is retrospective, as it merely caches the "effects" of past actions, whereas a MB system is prospective, in that it plans actions based on evaluating the future consequences of these actions with respect to one's goals. However, models of the task and the environment can be used for other functions. Here, we introduce a theory of MB retrospective inference. This theory addresses the frequent occurrence in which many of our actions are executed under conditions of state uncertainty, i.e. when important aspects of the state are latent. The theory proposes that we use our model of the world to resolve retrospectively, at least partially, an uncertainty operative at the very time actions were taken with implications for credit assignment. The MB system, therefore, does not focus solely on forming future action-plans, but has a function in shaping the impact of past experiences.

Our findings show that in the context of our task a MF system, which caches rewards obtained from choosing different objects without relying on a transition structure, can use retrospectively inferred knowledge about the ghost's nomination to selectively assign outcome credit to the relevant object. Indeed, we found that on uncertainty trials, MF learning was spared for the ghost-nominated object, i.e. it occurred with a rate that was similar to standard, no-uncertainty, trials. For the ghost-rejected object, on the other hand, learning was hindered. Note that credit was still assigned to the ghost-rejected object, a finding that is expected if some participants do not retrospectively resolve uncertainty, or if participants resolve it only part of the time. A striking aspect of our findings is that MF credit assignment discriminated in favour of the inferred ghost-nominated object, not only for an informative, inference-supporting, second outcome, but also for the non-informative first outcome when observers still maintain a 50–50% belief. An important question pertains to potential mechanisms that might account for these findings. We consider two, which we discuss in turn, namely delayed learning and a DYNA[17,27] architecture.

A delayed learning account rests on the idea that an MF-teaching prediction-error signal, based on total trial-reward, is calculated only at the end of uncertainty trials, after inference has already occurred. Notably, our task offered certainty that this uncertainty would be resolved later (i.e. after observing both outcomes). Thus, the MF system could in principle postpone a first-outcome credit assignment and "wait" for a MB inference triggered by observing a second outcome. Once a MB system infers the relevant object it can presumably increase a MF eligibility trace[33] for the inferred object and reduce it for the non-selected object, a process manifest as a higher MF learning rate for the inferred object. Many real-life situations, however, impose greater challenges on learners because it is unknown when, and even whether, uncertainty will be resolved. Indeed, there can often be complex chains of causal inference involving successive, partial, revelations from states and rewards. In such circumstances, postponing credit assignment could be self-defeating. Thus, it may be more beneficial to assign credit in real-time according to one's current belief state, and later on, if and when inferences about past states become available, enact "corrective" credit assignments adjustments. For example, upon taking an action and receiving a reward in an uncertain state, credit could be assigned based on one's belief state. Later, if the original state is inferred, then one can retrospectively deduct credit from the non-inferred state(s) and boost credit for the inferred state. Our findings pave the way for deeper investigations of these important questions and where it would be appealing to exploit tasks in which uncertainties arise, and are resolved, in a more naturalistic manner, gaining realism at the expense of experimental precision. More broadly, we consider much can be learned from studies that address neural processes that support MB-retrospective inference, how such inferences facilitate efficient adaptation, and how they contribute to MB-MF interactions.

Our findings also speak to a rich literature concerning various forms of MB-MF system interactions. In particular, the findings resonate with a DYNA architecture, according to which the MB system trains a MF system by generating offline (i.e. during inter-trial intervals) hypothetical model-based episodes from which an MF system learns as if they were real. Our findings suggest another form of MB-guidance, inference-based guidance whereby upon resolving uncertainty, an MB system indexes appropriate objects for MF credit assignment. One intriguing possibility arises if one integrates our findings with DYNA-like approaches. Consider a case where the MF system assigns credit equally to inferred and non-inferred objects online (i.e. during reward administration), but that MB inference biases the content of offline-replayed MB-episodes. For example, if the MB system is biased to replay choices of the inferred, as opposed to the non-inferred object (i.e. replay the last trial with its uncertainty resolved), then this would account for enhanced MF-learning for inferred relative to the non-inferred object. Future studies can address this possibility by manipulating variables that affect offline-replay such as cognitive load[17] or perhaps by direct examination of replay that exploits neurophysiological measures.

The importance of retrospective inference chimes with a truism that those who learn from history can also improve upon it. Consider, for example, a situation in which the ghost's nominations are biased towards the upper object rather than being uniform. In this case, inference about the ghost nominations would allow an agent to learn about the ghost's bias, which could subsequently be deployed in the service of better planning. A critical facet of our task, however, is that the MB system was provided with no incentive to perform retrospective inference about the ghost's nominee. Indeed, in our task the challenge faced by the MB system was to maintain accurate representations of the values of the various rooms and to calculate, based on the task's transition structure, prospective expected rewards for offered bandits. Because outcomes were fully-observed in uncertainty trials, their values could still be updated and the question of which object the ghost actually nominated was inconsequential. Put differently, retrospective inference was irrelevant with respect to either MB learning or future planning. We contend that the fact that an MB system still engaged in this effort-demanding process attests to its ubiquity and importance, and this is likely to be a "lower bound" on its extent. We would expect retrospective inference to be more substantial when it aids forthcoming MB planning, and this is an interesting question for further study. We acknowledge here theories that posit an intrinsic value of information[34–39], according to which information (which could be obtained by retrospective-inference) is rewarding in itself even when it lacks instrumental importance.

We conclude by noting that while there has been ample theoretical and empirical work on both dual RL systems, and on state uncertainty RL, these two literatures have mostly developed along distinct trajectories with minimal cross-talk. We believe that an

integration of these interesting fields can yield fundamental insights. The current work is but a first step in this project.

## Methods

**Participants.** Forty seven participants (29 female, 18 male) were recruited form the SONA subject pool (https://uclpsychology.sona-systems.com/Default.aspx?ReturnUrl=/) with the restrictions of having normal or corrected vision and being born after 1971. The study was approved by the University College London Research Ethics Committee (Project ID 4446/001). Subjects gave written informed consent before the experiment.

**Experimental procedures.** Participants were first familiarised with four pictures of objects and learned which pair of rooms each object opened (the pictures of the four objects were adopted from previous studies[40,41]). Each room was opened by two different objects and each object opened a unique pair of rooms. The mapping between objects and rooms was created randomly anew for each participant and remained stationary throughout the task. After learning, participants were quizzed about which rooms each object opened and about which object they would choose to open a target room. Participants iterated between learning and quiz phases until they achieved perfect quiz performance (100% accuracy and RT < 3000 ms for each question).

After learning participants played 16 practice standard bandit trials, to verify that the task was understood. These practice trials proceeded as described below with the sole difference that no time limit was imposed on a choice. They next played a single block of 72 standard bandit trials. On each trial, a pair of the four objects were offered for choice and participants had 2 s to choose one of these objects (left or right). Offered objects always shared one Common outcome. This defined four object-pairs, each presented on 18 trials, in a random order. Following a choice, the room Unique to the chosen object was opened and it was either empty (0 pt.) or included a treasure (1 pt.). Next, the room that was Common to both objects was opened, revealing it to be empty or with treasure. The reward probabilities of the four rooms evolved across trials according to four independent Gaussian-increment random walks with reflecting boundaries at $p = 0$ and $p = 1$ and a standard deviation of 0.025 per trial.

On completion of the first block participants were instructed about the uncertainty trials. On uncertainty trials participants were offered two disjoint object-pairs and were asked to choose the left or right pair. Objects within each pair always shared one Common outcome. Participants were instructed that a ghost would toss a fair and transparent coin and choose the vertically upper or lower object in their chosen pair, based on the coin-outcome. Once the ghost-nominated an object the trial proceeded as a standard trial, i.e. the two related rooms opened and treasures earned. Importantly, the room that was Common to the objects in the chosen pair was opened first while the room that was Unique to the ghost-nominated object was opened later. Following instructions, participant played a practice block consisting 18 trials, with each third trial being an uncertainty trial. During this practice block there was no time limit on choices. Following the practice trials, the test trials started. Participants played seven block of 72 trials each and short breaks were enforced between blocks. Choices were limited to 2 s and each third trial was an uncertainty trial. The 168 $3n + 1$ standard trials included 42 presentations of each of the four eligible object-pairs, in a random order. The 168, $3n + 2$ uncertainty trial included 84 repetitions of each of the two eligible pairings in a random order. Trials $3n + 3$ were defined according to their transition types relative to the choice on the preceding uncertainty trial. These 168 trials included 56 of each of the "repeat", "switch" of "clash" types in random order. A repeat trial presented the ghost-nominated object alongside its vertical counterpart, a switch trial presented the ghost-rejected object alongside its vertical counterpart and a clash trial presented the previously selected pair.

The task lasted between 90–120 min. Participants were paid £7.5 per hour plus a performance based bonus, which was calculated based on the total amount of earned treasure points.

**Data analysis.** One participant reported not feeling well and retired voluntarily from the experiment. Six other participants failed to pass the quiz within an hour and therefore did not start the task. The remaining 40 participants were the targets for the analysis.

**Model agnostic analysis.** Our model-agnostic analyses were conducted using logistic mixed effect models (implemented with MATLAB's function "fitglme") with participants serving as random effects with a free covariance matrix. For the MF-contribution analysis (Fig. 2a–e), we analysed only standard trials $n + 1$ that offered for choice the standard trial-$n$ chosen object. Our regressors C (Common outcome) and U (Unique outcome) coded whether trial-n outcomes were rewarding (coded as +0.5 for reward and −0.5 for non-reward), and the regressed variable REPEAT indicated whether the choice on the focal trial-$n + 1$ was repeated. PART coded the participant contributing each trial. The model, in Wilkinson notation, was: REPEAT~ C*U + (C*U|PART). For the MB-contribution analysis (Fig. 2f–n), we analysed only standard trials $n + 1$ that excluded from choice the standard trial-$n$ chosen object. The regressors C, U and PART were coded as in the previous analysis and one additional regressor P coded the reward probability of

the Common outcome (we centralized this regressor by subtracting 0.5). The regressed variable GENERALIZE indicated whether the choice on the focal trial-$n + 1$ was generalized. The model, in Wilkinson notation, was: GENERAL-IZED~C*P + (C*P|PART). We also tested an extended model GENERALIZED~ C*U*P + (C*U*P|PART) but found that none of the effects involving U were significant while the C*P effects supported the same conclusions.

The analyses that focused on MF learning on uncertainty trials considered standard $n + 1$ trials following an uncertainty $n$-trial (Fig. 4). The first analysis that examined learning for the ghost-nominated object (Fig. 4a) focused on "repeat" follow-ups, that is trials $n + 1$, that offered for choice the ghost-nominated object alongside the object from the previously non-selected pair that provided the previously inference-allowing outcome. A choice repetition was defined as a choice of the ghost-nominated object. We used the model REPEAT~ N*I + (N*I |PART), where N and I coded outcomes (−0.5, 0.5) obtained on trial-$n$ from the Non-informative and the Informative room, respectively. The second analysis that examined learning for the ghost-rejected object (Fig. 4b) focused on "switch" follow-ups, that is trials $n + 1$, that offered for choice the ghost-rejected object alongside an object from the previously non-selected pair that shared an outcome with the previously ghost-rejected object. A choice generalization was defined as a choice of the ghost-rejected object. We used the model GENERALIZE~ N*I + (N*I|PART). Finally, the third analysis that compared first-outcome learning for the ghost-nominated and -rejected objects (Fig. 4c) focused on "clash" follow-ups, that is trials $n + 1$, that offered for choice the ghost-nominated and ghost-rejected objects. A choice repetition was defined as a choice of the ghost-nominated object. We used the model REPEAT~N*I + (N*I|PART).

**Computational models.** We formulated two hybrid RL models to account for the series of choices for each participant. In both models, choices are contributed by both the MB and MF systems and they differed only in how the MB system operates.

In both models, The MF system caches a $Q^{MF}$-value for each object, subsequently retrieved when the object is offered for choice. When a pair of objects is offered for choice (on uncertainty trials), the MF pair-value was calculated as the average of constituent object's MF-values:

$$Q^{MF}(\text{pair}) = \frac{Q^{MF}(\text{object 1}) + Q^{MF}(\text{object 2})}{2} \quad (4)$$

On standard trials, the total reward (in points) from outcomes is used to update the $Q^{MF}$-value for the chosen object, based on a prediction error (with a free learning rate parameter $lr_{\text{standard}}$):

$$Q^{MF}(\text{object}) \leftarrow Q^{MF}(\text{object}) + lr_{\text{standard}} * (\text{total reward} - Q^{MF}(\text{object})) \quad (5)$$

On uncertainty trials, the MF system updates the $Q^{MF}$-values for both the ghost-nominated and -rejected objects with free learning rate parameters $lr_{\text{ghost-nom}}$ and $lr_{\text{ghost-rej}}$, respectively:

$$Q^{MF}(\text{selected}) \leftarrow Q^{MF}(\text{selected}) + lr_{\text{ghost-nom}} * (\text{total reward} - Q^{MF}(\text{selected})) \quad (6)$$

$$Q^{MF}(\text{rejected}) \leftarrow Q^{MF}(\text{rejected}) + lr_{\text{ghost-rej}} * (\text{total reward} - Q^{MF}(\text{rejected})) \quad (7)$$

Our two models postulated alternative "room-value learning" and "object-value learning" formulations for an MB system. We describe these models in turn. In the room-value learning formulation (the model variant presented in the main text), the MB system maintains $Q^{MB}$-values for the four different rooms. During choice on standard trials the $Q^{MB}$-value for each offered object is calculated based on the transition structure:

$$Q^{MB}(\text{object}) = Q^{MB}(\text{room 1}) + Q^{MB}(\text{room 2}) \quad (8)$$

During uncertainty trials the $Q^{MB}$-value of each pair is calculated based on the average constituent values:

$$Q^{MB}(\text{pair}) = \frac{Q^{MB}(\text{object 1}) + Q^{MB}(\text{object 2})}{2} \quad (9)$$

Following a choice, the MB system updates the $Q^{MB}$-values of each observed rooms:

$$Q^{MB}(\text{room}) \leftarrow Q^{MB}(\text{room}) + lr_{MB} * (\text{reward} - Q^{MB}(\text{room})) \quad (10)$$

where $lr_{MB}$ is an MB learning rate parameter.

Alternatively, in the object-value learning formulation, during the choice phase the MB system retrieves the values of the choice-objects, much like the MF system. Additionally, MB learns during reward administration the values of objects rather than rooms, but unlike MF, it takes into account the task's transition structure. For

the chosen object, i.e. the object that was chosen by the participant on standard trials or by the ghost on ghost trials, a "full update" is performed:

$$Q^{MB}(object) \leftarrow Q^{MB}(object) + lr_{MB} * (total\ reward - Q^{MB}(object)) \quad (11)$$

For each of the two other non-chosen objects, each of which provides only one of the two experienced room-outcomes, a "half update" was performed based on the relevant room.

$$Q^{MB}(object) \leftarrow Q^{MB}(object) + lr_{MB} * (room\ reward - 0.5 * Q^{MB}(object)) \quad (12)$$

For example (see Fig. 1a), when the key was chosen (either by the participant or the ghost) a full update was performed for the key, and half updates based on the brown and green rooms, respectively, were performed for the phone and the stove.

In both models, when a pair of objects are offered for choice on standard trials the net Q value of each object is calculated as

$$Q_{net}(object) = w_{MB} * Q^{MB}(object) + (1 - w_{MB}) * Q^{MF}(object) + p * 1_{last\ chosen} \quad (13)$$

Where $w_{MB}$ is a free parameter (between 0–1) that quantifies the relative contribution of the MB system ($1 - w_{MB}$ is therefore the relative MF contribution), $p$ is a free perseverance parameter, which quantifies a general tendency to select the object that was last selected, and $1_{lastchosen}$ indicates whether the focal object was selected on the previous trial. On uncertainty trials the value of each offered pair is calculated similarly as:

$$Q_{net}(pair) = w_{MB} * Q^{MB}(pair) + (1 - w_{MB}) * Q^{MF}(pair) + p * 1_{last\ chosen} \quad (14)$$

where here, $1_{lastchosen}$ indicates whether the focal pair includes the previously selected object. The $Q_{net}$ values for the two choice options (objects or object-pairs) are then injected into a softmax choice rule with a free inverse temperature parameter $\beta > 0$ so that the probability to choose an option is:

$$Prob(option) = \frac{e^{\beta * Q_{net}(option)}}{e^{\beta * [Q_{net}(option) + Q_{net}(other\ option)]}} \quad (15)$$

MF $Q^{MF}$-values where initialized to 1 for each object and MB $Q^{MB}$-values were initialized to 0.5 for each room.

We also formulated two pure MF models with either accumulating or replacing eligibility traces[33] to test whether these mechanisms, rather than MB inference guided learning, could account for our findings. In these models, the MB contribution was silenced by setting $w_{MB} = 0$ and removing $lr_{MB}$ from the model. These models included a single free learning rate parameter for the MF system, $lr_{MF}$, and a free eligibility trace (decay) parameter, denoted $\lambda$. For each of the four objects we maintained throughout the series of trials, an eligibility trace, $e(object)$. At the beginning of the experimental session, these traces were initialized to 0. At the end of each trial all four eligibility traces decayed according to

$$e(object) \leftarrow \lambda * e(object) \quad (16)$$

Immediately after a choice was made on a standard trial the eligibility trace of the chosen object was updated. In accumulating traces model we set

$$e(chosen\ object) \leftarrow 1 + e(chosen\ object) \quad (17)$$

And in the replacing traces model we set

$$e(chosen\ object) \leftarrow 1 \quad (18)$$

On ghost trials the eligibility traces for both objects in the chosen pair were thus updated. Finally, following reward administration the value of each of the four objects was updated. For accumulating eligibility traces we set

$$Q^{MF}(object) \leftarrow Q^{MF}(object) + (1 - \lambda) * lr_{MF} * e(object) * (total\ reward - Q^{MF}(object)) \quad (19)$$

And for replacing trace:

$$Q^{MF}(object) \leftarrow Q^{MF}(object) + lr_{MF} * e(object) * (total\ reward - Q^{MF}(object)) \quad (20)$$

In sum, in the eligibility-trace models, the sequence of model calculation during a trial consisted of subjecting all eligibility traces to a decay, making a choice, increasing the eligibility trace(s) for the chosen object(s), obtaining outcomes (rewards or non-rewards) and updating the values of all four objects.

**Model fitting and model comparison**. We fit our models to the data of each individual, maximizing the likelihood (ML) of their choices (we optimized likelihood using MATLAB's 'fmincon', with 200 random starting points per

participant). Each of our two full hybrid models, which allowed for contributions from both an MB and an MF system, served as a super-model in a family of nested sub-models: the room-value learning and the object-value learning families. Each family consisted of four sub-models: The first, a pure MB model, constrained the model-based relative contribution to 1, $w_{MB} = 1$, and the learning rates of the MF system to 0, $lr_{standard} = lr_{ghost-nom} = lr_{ghost-rej} = 0$. The second, a pure MF-action model, constrained the MF relative contribution to choices to 1, $w_{MB} = 0$, and the MB learning rate to 0, $lr_{MB} = 0$. Note, however, that the MB system was still able to guide MF learning through inference. The third, 'non-inference' sub-model constrained equal learning rates for the ghost-nominated and -rejected objects $lr_{ghost-nom} = lr_{ghost-rej}$. The fourth, 'no-learning for ghost rejected object' constrained the learning rate of the ghost-rejected object to 0: $lr_{ghost-rej} = 0$. The best-fitting parameters for the super-model are reported in Supplementary Table 1.

We next conducted, for each family separately, a bootstrapped generalized likelihood ratio test (BGLRT[31]) for the super-model vs. each of the sub-model separately (Fig. 5). In a nutshell, this method is based on the classical-statistics hypothesis testing approach and specifically on the generalized-likelihood ratio test (GLRT). However, whereas GLRT assumes asymptotic Chi-squared null distribution for the log-likelihood improvement of a super-model over a sub-model, in BGLRT these distributions are derived empirically based on a parametric bootstrap method. In each of our model comparison the sub-model serves as the H0 null hypothesis whereas the full model as the alternative H1 hypothesis.

For each participant, we created 1001 synthetic experimental sessions by simulating the sub-model with the ML parameters on novel trial sequences, which were generated as in the actual data. We next fitted both the super-model and the sub-model to each synthetic dataset and calculated the improvement in twice the logarithm of the likelihood for the full model. For each participant, these 1001 likelihood-improvement values served as a null distribution to reject the sub-model. The $p$-value for each participant was calculated based on the proportion of synthetic dataset for which the twice logarithm of the likelihood-improvement was at least as large as the empirical improvement. Additionally, we performed the model comparison at the group level. We repeated the following 10,000 times. For each participant we chose randomly, and uniformly, one of his/her 1000 synthetic twice log-likelihood super-model improvements and we summed across participant. These 10,000 obtained values constitute the distribution of group super-model likelihood improvement under the null hypothesis. We then calculated the $p$-value for rejecting the sub-model at the group level as the proportion of synthetic datasets for which the super-model twice logarithm of the likelihood improvement was larger or equal to the empirical improvement in super-model, summed across-participants.

Additionally, we compared between the room-value learning and the object-value learning full models using the parametric bootstrap cross-fitting method (PBCM[42]). For each participant, and for each of the two model-variants, we generated 100 synthetic experimental sessions by simulating the model using the ML parameters on novel trial sequences (which were generated as in the experiment). We then fit each of these synthetic datasets with both models. Next we repeated the following 10,000 times, focusing on data that was generated by the room-value learning model. For each participant we chose randomly and uniformly one of his/her 100 synthetic datasets and calculated twice the log-likelihood difference between the fits of the room-value learning model and the object-value learning models. These differences were averaged across participants. Thus, we obtained 10,000 values that represent the distribution of the group twice log-likelihood difference for data that is generated by the room-value learning model. Next, we repeated the same steps but this time for synthetic data that was generated by the object-value leaning model, to obtain a distribution of the group twice log-likelihood difference for data that is generated by the object-value learning model. An examination of these two distributions (Supplementary Fig. 5) showed that each model provided a better fit for the group data in terms of likelihood when it is the generating model. We thus set log-likelihood difference of 0 as the model-classification criterion with positive difference supporting the room-value learning model and negative values supporting the object-value learning model. Finally, we averaged twice the log-likelihood difference for the empirical data across participants, to obtain the empirical group difference. This difference was 4.71, showing that the room-value learning model provides a superior account for the group data.

**Model simulations**. To generate model predictions (Fig. 2, Supplementary Figs. 1–3), we simulated for each participant, 25 synthetic experimental sessions (novel trial sequences were generated as in the actual experiment), based on his or her ML parameters obtained from the corresponding model fits (the models are described above). We then analysed these data in the same way as the original empirical data (but with datasets that were 25 times larger, as compared to the empirical data, per participant).

**Comparing MF learning rates**. We compared the estimated MF learning rates for standard-chosen, ghost-nominated and ghost-rejected objects, using a linear mixed effect model (implemented with MATLAB's function "fitglme") with participants serving as random effects with a free covariance matrix (Fig. 6a). Regressors GS (ghost-nominated) GR (ghost-rejected) indicated whether the learning rate corresponded to the ghost-nominated object and to the ghost-rejected object, respectively. The regressed variable LR was the estimated learning rate. PART

coded the participant. The model, in Wilkinson notation, was: LR~GS+GR + (GS + GR|PART). We followed-up with an *F*-test that rejected the hypothesis that both GS and GR main effects were 0, indicating that the three learning rates are different. We next contrasted all three learning rates pairs.

**Correlation between MB and MF preferential learning**. Based on the ML parameters of the full models, we calculated (Fig. 6b) the across-participants correlation between model-basedness ($w_{MB}$) and the MF preferential learning for the ghost-nominated object ($lr_{ghost-nom} − lr_{ghost-rej}$). The significance of this correlation was calculated based on a permutation test in which $w_{MB}$ was shuffled across participants.

We note, however, that the empirical data showed an increase in participant heterogeneity with respect to preferential learning as a function of model-basedness. This is evident in Fig. 6b in both the differences between participants and in the increase of individual error bars as model basedness increases. This occurs because the MF learning rates exert a weaker influence on performance as "model-basedness" increases (and the relative MF contribution decreases) and hence, learning-rate estimation noise increases. One caveat pertaining to the above permutation test is that it fails to control for this increasing heterogeneity pattern, as this pattern will vanish in shuffled data. To address the possibility that this pattern generated a spurious positive correlation we conducted a control test (Supplementary Fig. 7). We parametrically bootstrapped (using model simulations) 1000 synthetic experimental sessions for each participant's data based on the ML parameters from the non-inference model in which, preferential learning for the ghost-nominated object is absent ($lr_{ghost-nom} = lr_{ghost-rej}$). We next fitted each of these synthetic datasets with the full model to obtain estimates of model-baseness and preferential learning. Next we repeated the following 100,000 times: We chose for each participant randomly the fitting parameters obtained for one of his/her 1000 synthetic datasets and we calculated the group correlation between model-baseness and preferential learning. Because this correlation is calculated for data that featured no correlation, the 100,000 values comprise a null distribution for the expected correlation. The significance value of the empirical correlation was calculated as the proportion of samples in the null distribution that were larger than the observed correlation.

**Error bars for the MF preferential learning effect**. We calculated the individual error bars for the difference between MF-learning rates for the ghost-nominated and ghost-rejected objects (Fig. 6b) as follows. For each participant, we generated 100 synthetic experimental sessions by bootstrapping his/her data based on the ML parameters of the full room-value learning model. We then fitted the full model to each of these synthetic datasets and calculated the difference between the ghost-nominated and ghost-rejected learning rates. This provided an estimate of the expected distribution of the learning rate-difference, had we been able to test a participant multiple times. Next we found the densest (i.e. narrowest) interval that contained 50% of the mass of this distribution, conditional on the interval including the empirical learning rate difference.

**Reporting summary**. Further information on experimental design is available in the Nature Research Reporting Summary linked to this article.

## Data availability

The data that support the findings of this study and data analysis code have been deposited in the Open Science Framework (OSF) and are available in the following link: [https://osf.io/8j7yf/?view_only=8362bdb2672643de98daaa8e509aae30].

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

## Acknowledgements

P.D. was on a leave of absence at Uber Technologies during part of time that this work was being carried out. We thank Eran Eldar for his helpful comments on the MS. R.M., M.K. and R.J.D. were funded by the Max Planck Society, Munich, Germany, URL: https://www.mpg.de/en, Grant number: 647070403019. R.J.D. was also funded by the Wellcome Trust, URL: https://wellcome.ac.uk/home, Grant number/reference: 098362/Z/12/Z. P.D. was funded by the Gatsby Charitable Foundation and by the Max Planck Society.

## Author contributions

R.M., P.D. and R.J.D. conceived the study. R.M. designed the experiment. R.M. programmed the experiments. R.M. performed the experiments. R.M. developed the models. R.M. analysed the data. R.M., P.D. and M.K. interpreted the results. R.M. drafted the manuscript. All authors wrote the manuscript.

## Additional information

**Competing interests:** The authors declare no competing interests.

