## [Peer Review File · Nature Communications]

Reviewers' comments:

Reviewer #1 (Remarks to the Author):

This manuscript is based on a dichotomy well established in the reinforcement learning literature between model-free (MF) and model-based learning (MB). However, while well-studied, the authors take a completely new perspective on this duality. Rather than focusing on a role of the MB system in forward planning, they suggest a “retroactive” influence of the MB system on credit assignment in the MF system. According to this perspective, the MB information can disambiguate state uncertainty to allow a model-free learner to assign credit for a reward to the choice that caused that reward. This idea is novel and thought provoking. The authors use a cleverly designed task and elegant analysis approaches to investigate this proposed mechanism. Logistic regression approaches, analysis of choice repetition probabilities and parameters derived from fitting RL models appear to support their hypotheses. All in all, I anticipate that this manuscript will be an important novel contribution to the field.

(1) My main issue is with how the results are presented. Reading through the manuscript, everything sounded straightforward and easy to understand. I had a harder time working through figures 2 and 4 however. My personal preference would be to use bar instead of line plots for the effects of common/unique rewards/non-rewards. Also, while I appreciate the authors reasoning to reduce the plots and only show the rooms associated with the objects, I guess I would prefer if you could also show the objects that were chosen/non-chosen. I always had to revisit figures 1 and 3 to back-infer which actual object choice was being referred to in the example.

(2) Results in Fig 2: The logistic regression results mentioned in the text (p. 10/11) show an effect of both the common and the unique room outcome to repeat the same object choice on the next standard trial. This does not seem to be reflected in the choice repetition probabilities in Fig 2 – or at least it is not evident to me: choice repetition probability is not > 0.5 when the common room is rewarded \neg (unless there is also reward on the unique room). Would one not expect $p(\text{repeat}) > 0.5$ when the common room outcome is assigned to the chosen object in a model-free fashion? I think this is important to clarify, since it seems a key point of the manuscript that subjects do indeed employ model-free learning (modelling results later on strongly suggest they do).

On the other hand, repetition probability is > 0.5 in both cases for reward on the unique room - whether or not the common room is rewarded (with even higher repetition probability when the

common room too is rewarded). This also appears in line with the logistic regression results, where coefficients for the unique room outcomes are higher than those for common room outcomes. Would this not argue for model-based contribution?

(3) A similar question arises in figure 4. If the MF system assigns credit to the ghost-chosen object, then reward on the unique room should increase the probability that it will be chosen again on trial $n+1$. Again, $p(\text{repeat})$ does appear to be larger than 0.5 when the common room is non-rewarded (or is it? It is hard to tell from the figure). By contrast, when the common room is rewarded, $p(\text{repeat})$ is much larger than 0.5, both when the unique room is rewarded and when it is not (albeit more so when it is). Could you please clarify?

(4) I could not follow the reasoning underlying figure 4B (p.17):

“From the perspective of the MF system, however, if credit was assigned to the ghost-rejected object on trial n , then a unique-reward, as compared to non-reward, should increase the generalization probability.”

Why? In the example shown, the unique brown room disambiguates state uncertainty, it is clear that the ghost-chosen object was the key. If the MB-system is indeed used to guide credit assignment retroactively, the credit for the unique room should be entirely assigned to the choice of “key”, shouldn’t it?

Modelling results were all very clear and intuitive.

Minor

Please decide to use either „unique“ or „uncommon“ throughout

Reviewer #2 (Remarks to the Author):

This paper presents a cleverly designed (though devilishly complicated) experiment designed to reveal how the resolution of state uncertainty can retrospectively assign credit to past actions. First, the authors establish that choices are to some extent controlled by a "model-free" system that learns to select actions based on past reinforcement in particular states. They then establish that the model-free system can assign credit to actions based on later events that resolve state uncertainty. Because this uncertainty resolution can only be accomplished by a "model-based" system, they reason that their findings reflect a cooperative synergy between model-free and model-based systems.

I found the paper overall well-written and compellingly argued. I had to re-read some sections multiple times because the experimental design is quite intricate. I'm not sure if much can be done to improve that aspect of the paper, because the intricacy seems necessary to test for these subtle effects.

Major comments:

It's somewhat paradoxical that the authors find a correlation between the degree of model-based control and the effect of uncertainty resolution on retrospective credit assignment. On the one hand, I understand the logic of this analysis to be that the retrospective credit assignment is only possible through the operation of a model-based system. On the other hand, if a subject is purely model-based, then they wouldn't show the effect at all. Eyeballing Figure 5C, I think there may actually be some evidence for this: there are a few subjects with a high degree of model-based control who show weaker, or even reversed retrospective credit assignment effects. This might be worth investigating further.

Another substantive issue is that the modeling is essentially descriptive (different learning rates for different trial types). The authors don't provide a plausible mechanistic account of how resolution of uncertainty contributes to credit assignment. I don't think this is crucial to making the points in their paper, but it might help readers better understand more explicitly what computations would produce these results.

As the authors know well, the idea that model-based inference guides model-free learning has been pursued in a number of other papers. Some of this literature is already cited (Lak et al., 2017; Starkweather et al., 2018) but not the seminal theoretical work on POMDPs (Kakade & Dayan, 2002; Daw, Courville & Touretzky, 2006; Rao, 2010) nor some of the relevant experimental work (Sarno et al., 2017; Babayan et al., 2018). The citations don't have to be exhaustive, but I thought they were a bit short-shrifted.

Minor comments:

abstract: "an environments" -> "an environment"

p. 4: "and calculates the expected rewards" -> "and calculate the expected rewards"

p. 4: "an agents'" -> "an agent's"

p. 10: I think it would help the reader if the authors unpack what they mean by "canceling out".

p. 14: "unique, inference allowing, outcome" misplaced commas

p. 15: "objected" -> "object"

references: the Keramati et al. (2017) citation should be Lak et al. (2017)

Reviewer #3 (Remarks to the Author):

In this work, Moran Keramati Dayan Dolan propose a novel theory of goal-directed (model-based; MB) retrospective-inference to resolve uncertainty in the environment that prevail when actions have been taken so as to guide a habitual (model-free; MF) credit-assignment. They moreover experimentally test this theory using a novel task, where there is uncertainty about chosen bandits (i.e. lotteries). They find that when participants' momentary uncertainty about which bandit had generated an outcome was resolved by provision of subsequent information, participants preferentially assigned credit to the bandit they retrospectively inferred was responsible for an experienced outcome, which suggests evidence for MB retrospective inference.

Overall, the task is well designed, addresses an important question, and the results are very interesting, but would nevertheless deserve a few more control analyses. More importantly, the

interpretation of the results (that they would be a hallmark of MB retrospective inference + MF credit assignment) is not yet fully convincing. The authors should control for alternative models such as a pure MF system with eligibility traces, a hybrid MF+MB model without retrospective inference, or a pure MB system learning the values of objects rather than the values of rooms (but still maintaining task structure information linking objects and rooms) which could show the common outcome effect by rewarding the two objects at outcome 1 and then would reinforce the ghost-selected object but not the ghost-rejected object at outcome 2. This model would predict that a third object associated to the same second room would also be reinforced, namely the telephone in the example of Figure 3 because it is also associated to the brown room. I suggest below additional analyses of the present data on other types of trials to help disentangle between alternative models and further illustrate evidence for either MF or MB. Some control analyses for model fitting are also suggested.

MAIN ISSUES

I- The result in Figure 4C is very nice and striking. Nevertheless, an important question is whether alternative models could also account for these results. First, the presentation of the task structure in Figure 1B suggests that the objects remain on the screen with the rooms during outcomes 1 and 2. Thus a hybrid MF+MB model without retrospective inference (where the MF reinforces all presented objects while the MB waits for the two outcomes and the two room presentations to know which object is associated to the two rooms and thus only then reinforces this object (rather than reinforcing room values), thus without retrospective-inference) seems a good candidate. Any choice made by this model would be a weighted sum of MF and MB and might thus produce all effects reported here. The authors could also test whether a pure MF model with eligibility traces (so that any positive outcome received not only reinforces the last selected object but also reinforces a little bit the second to last selected object) could account for this result. This would control if the observed effect is really a hallmark of retrospective inference or whether it could also be explained in terms of MF eligibility traces. The latter would moreover be consistent with the credit assigned during "switch trials" from the unique outcome to the ghost-rejected object. In contrast, the present interpretation for this result, "that an MF system assigns credit to the ghost-rejected object" while the MB system would be responsible for the retrospective inference, does not sound very parsimonious and forces the authors to refer to "MF credit-assignment" throughout the manuscript without really proving it, which I further discuss below. The present full model is even more problematic that it does not predict the MF ghost credit-assignment contrast (Figure S2E), in contrast to the experimental data (Figure 4D). In relation to this, I don't understand why the authors can write that they "validated our analysis-approach using model simulations (see Supplemental Material Fig.S2)" in line 348 while Figure S2E does not reproduce the experimental results with the full model.

II- As I understand Figure 2, it is intended to illustrate some predictions of the model and then compare to subjects' behavior. In the first place, it could be merged with Figure S1, showing (A) the

MF-contribution trials, (B-D) model predictions and then (E) the subjects' behavior. Then I think it would also be great to add a second line to Figure 2 (panels F-J) to highlight, symmetrically, some MB-contribution trials: cases where the same object is not presented at trial $n+1$, but rather an object that shares one common room (in this example, the stove or the phone) against an object that shares no room (the light bulb), and show how frequently MF and MB chose the latter as a function of C-Non/C-Rew and whether the two different rooms (D1 and D2) of the light bulb were both rewarded, both unrewarded, or only one rewarded. These additional panels would show a significant difference between blue (C-Non) and red (C-Rew) curves for MB but not for MF, which I think would further clarify different predictions of the two systems.

III- I am not totally convinced by the authors in lines 356-357 that "MB contribution predicts no effect of the trial n common outcome due to cancelling out", because one could think of a slightly different MB model where the model learns the value of objects rather than the value of rooms, still using task structure information (as written above). The model would make different predictions than that of the present model. So the point raised by the authors does not seem as a general MB prediction, but rather specific to the MB that they are using.

IV- To further disentangle MF and MB predictions, and similarly to the suggestions above for Figure 2, I would suggest adding another type of trials in Figure 4: uncertainty trials n followed by trials $n+1$ with a choice between an object associated to none of the rooms and an object associated to the unique room but not the common one (light bulb against telephone in the example of Figure 4). A preference for the latter would indicate that during outcome 2, credit was not assigned only to the object which is associated to the two rooms presented at outcome 1 and outcome 2 (the key in the example of Figure 4), but rather to all objects associated to the second room (key and telephone are both associated with the brown room).

V- A word about the terminology. From the end of the introduction and throughout the manuscript, the authors repeatedly employ the expressions "MF credit-assignment", "MF assignment of credit", "MF learning". First, this sounds a bit strange when it is associated with MB inference or with the MB system in the same sentence (for instance lines 96-97) because the reader does not yet know that the authors intend to later present results to specifically address MF learning, and thus does not understand at this stage why the credit assignment should necessarily be MF rather than simply be a later preference of the MF system towards the inferred state. One may think of the DYNA architecture by Richard Sutton, which is cited by the authors. But it is only at the end of the manuscript that the authors really discuss this. Second, the authors continue to use the same expressions when presenting results that focus on MB retrospective-inference, while at these other moments it does not really matter whether the credit assignment is done in a MF way or a MB way. For instance, lines 149-152, lines 228-229, lines 247-248, lines 261-263, line 309. So, to facilitate reading, I would suggest to replace all occurrences of "MF learning", "MF credit-assignment", "MF assignment of credit" by "learning", "credit-assignment", "assignment of credit", respectively. Finally, and still with respect to the same issue, the reinforcement arrow for the key in Figure 3 is

thicker than that of the stove not only after Outcome 2 but also after Outcome 1, while at this stage of the reading, the reader does not know that the subjects showed this effect (Figure 4) and rather think that the arrows should be equally thick because the two objects should be equally rewarded. A note in the figure legend would be useful.

VI- Individual differences. In lines 403 and 406 respectively, 14 and 10 participants respectively are both minorities of the total of 40 participants. Could the authors be more explicit about which alternative models win for the other participants at the individual-level?

VII- Model fitting. Could the authors verify in which percentage of the 20 starting points per model and per participant the gradient descent performed by `fmincon` converged to the same point? If this percentage is small, the authors may consider to increase the number of starting points in order to increase the chances of converging to the global optimum.

SECONDARY ISSUES

Does this really make sense: "vertical counterpart from the non-selected pair (this is the object that the ghost would have chosen had the participant chosen the other pair)"? How can the subject know what the ghost would have chosen from the non-selected pair? Why would one object be more likely chosen by the ghost within that non-selected pair?

Please keep a consistent terminology, for instance "unique outcome" rather than "second-outcome" in Figure S2 legend.

In line 381, the authors should have written "ghost_{rej}" as subscript of the third MF learning rate.

In lines 674-677 the title of the article is repeated twice. Moreover, the first author is Armin Lak, not Mehdi Keramati. Finally, this reference is repeated in lines 740-742.

We are very grateful to the reviewers for their careful reading of, and helpful comments, on our paper. We have addressed all the points raised. We detail our responses below. Since they occasioned a large number of changes to the paper, we only copy the most substantial edits into this reply. However, note that all the material that we changed is marked in yellow in the main text.

Reviewer #1 (Remarks to the Author):

This manuscript is based on a dichotomy well established in the reinforcement learning literature between model-free (MF) and model-based learning (MB). However, while well-studied, the authors take a completely new perspective on this duality. Rather than focusing on a role of the MB system in forward planning, they suggest a “retroactive” influence of the MB system on credit assignment in the MF system. According to this perspective, the MB information can disambiguate state uncertainty to allow a model-free learner to assign credit for a reward to the choice that caused that reward. This idea is novel and thought provoking. The authors use a cleverly designed task and elegant analysis approaches to investigate this proposed mechanism. Logistic regression approaches, analysis of choice repetition probabilities and parameters derived from fitting RL models appear to support their hypotheses. All in all, I anticipate that this manuscript will be an important novel contribution to the field.

We thank the reviewer for a positive, insightful and constructive evaluation of our findings.

(1) My main issue is with how the results are presented. Reading through the manuscript, everything sounded straightforward and easy to understand. I had a harder time working through figures 2 and 4 however. My personal preference would be to use bar instead of line plots for the effects of common/unique rewards/non-rewards. Also, while I appreciate the authors reasoning to reduce the plots and only show the rooms associated with the objects, I guess I would prefer if you could also show the objects that were chosen/non-chosen. I always had to revisit figures 1 and 3 to back-infer which actual object choice was being referred to in the example.

Thank you for these suggestions. We apologise for the complexity of the figures. In line with the suggestions we have revised the ones mentioned to include the objects (so we hope that no back reference to Fig. 1 is necessary). We tried using bar plots instead of line plots, but found that the latter were less cluttered, and so we consider these are a bit easier for our target audience to understand. We therefore prefer to keep that format. We hope that with the clarifications provided below the reviewer and readers will find the plots clearer.

For example, here is what Fig. 2B looks like in both formats

(2) Results in Fig 2: The logistic regression results mentioned in the text (p. 10/11) show an effect of both the common and the unique room outcome to repeat the same object choice on the next standard trial. This does not seem to be reflected in the choice repetition probabilities in Fig 2 – or at least it is not evident to me: choice repetition probability is not > 0.5 when the common room is rewarded (unless there is also reward on the unique room).

We thank the reviewer for these questions. The reviewer brings up a number of issues which we will clarify in turn, and also in answer to the next question.

First, considering the logistic regression, the positive effect of the common room reflects the fact that the C-rew (red) line is above the C-non (blue) line for both values of the unique outcome (U-non and U-rew). The predicted probabilities from the regression model are influenced by all regression coefficients: the intercept, the effects of the common and unique rooms and by their interaction – and indeed the reviewer’s intuition that the effect of the unique room is greater than that of the common room is quite correct. However, these probabilities cannot be compared against a nominal baseline probability of 0.5.

For a worked example: in the relevant regression model, coding reward and non-reward as +0.5 and -0.5 respectively, the coefficients were:

coefficient:	value
beta_intercept	0.41
beta_common	0.98
beta_unique	2.03
beta_common_unique_interaction	0.46

Thus, the predicted repetition probability (in logistic units) for the case in which the common room is rewarded and the unique room is unrewarded is

$$\text{beta_intercept} + 0.5 \cdot \text{beta_common} - 0.5 \cdot \text{beta_unique} - 0.25 \cdot \text{beta_interaction} = -0.23.$$

The fact that this value is negative corresponds to a nominal repetition probability lower than 0.5.

The repetition-probability for the mirror case for which the common room is unrewarded, but the unique room is rewarded, can be calculated similarly as

$\text{beta_intercept} - 0.5 * \text{beta_common} + 0.5 * \text{beta_unique} - 0.25 * \text{beta_interaction} = 0.82.$

and thus a repetition probability which is $> .5$.

Thus, the regression model is able to predict probabilities that were higher and lower than $.5$, in these two cases, respectively, for the very reason that $\text{beta_unique} > \text{beta_common}$.

In conclusion, there is no inconsistency between the results of the logistic regression and how the plotted probabilities relate to $.5$.

Would one not expect $p(\text{repeat}) > 0.5$ when the common room outcome is assigned to the chosen object in a model-free fashion? I think this is important to clarify, since it seems a key point of the manuscript that subjects do indeed employ model-free learning (modelling results later on strongly suggest they do).

Importantly, we highlight that the data in the figure concerned do not reflect a pure MF contribution. Rather, the empirical data reflect the non-linear aggregation of a multitude of influences including, but not limited to, the MF contribution. These additional influences are:

- 1) A contribution from a MB system,
- 2) A tendency towards perseveration (which could be either positive or negative) and
- 3) The historical values of the objects as determined by past reward events that occurred before trial n .

The impact of these terms implies that a comparison should not be made with a probability of 0.5 to assay whether or not there are MF contributions to behaviour.

To elaborate, consider the predictions of a pure MB agent: when the common room is rewarded and the uncommon room is unrewarded, a pure MB agent can predict a $p < .5$ repetition probability. The reason is that the common reward is “cancelled out” from MB calculations on trial $n+1$, whereas the unique non-reward decreases the value of the repetition object relative to the object that is the alternative on trial $n+1$. In support of this argument, when we simulate the behaviour of a pure MB agent based on the best-fitting parameters of our full model, but knocking out contributions from both the MF system (setting $w_{\text{MB}}=1$) and perseveration (setting $\text{Perseveration}=0$), we found that the model predicted a repetition probability < 0.5 .

In sum, the repetition probability reflects the aggregation of several influences, some acting to increase it but others acting to decrease it. The predicted probability will therefore depend on the specific parameters that determine how these influences combine. Critically, a repetition probability < 0.5 is consistent with contributions from a MF system.

On the other hand, repetition probability is > 0.5 in both cases for reward on the unique room - whether or not the common room is rewarded (with even higher repetition probability when the common room too is rewarded). This also appears in line with the logistic regression results, where coefficients for the unique room outcomes are higher than those for common room outcomes.

As explained above, this is indeed correct!

Would this not argue for model-based contribution?

Thank you for this great suggestion/question. The somewhat surprising answer is no! A pure MF contribution to choices can also predict a greater influence for the unique compared to the common room. This is evident, for example, in the revised Fig. 2D, which shows the result of eliminating the influences that MB exerts on choices. Therefore, a larger unique influence does not implicate the contributions of a MB system.

The explanation for this is somewhat subtle, so we attempt here to provide the intuition. It arises from an impact of the autocorrelation in the random walks that generate the reward probabilities for each room on the sequence of MF values of the objects that lead to those rooms.

Thus, for instance, the green room will alternate between temporally extended epochs when it is either better or worse than its average.

Thus, consider the example in Fig 2A. The key and the stove share a common green room outcome. When the trial n choice of the key is rewarded (vs. unrewarded) because of the common green room, it is more likely that the green room is currently good. But this means that the MF value of the stove is also more likely to be higher on trial $n+1$ as the autocorrelation in the value of the green room implies that it is likely to have benefited from green-room rewards on recent past choices before trial n . It follows that the green room outcome on trial n (rewarded vs. unrewarded) correlates positively with the MF Q-value of the stove on trial $n+1$.

We stress that this correlation cannot be due to a direct causal influence of trial n events on the MF stove value on trial $n+1$ (because the stove was not chosen on trial $n+1$). Rather, it is due to the autocorrelative structure of the reward sequence. In any event, this means that when the green room is rewarded (vs. unrewarded) on trial n , the key has to compete on trial $n+1$ for choice with a more valued stove. This mitigates the effect of the trial n green outcome on MF choice repetition. The effect of the unique brown room, on the other hand, is not subject to a similar mitigation, because it is not an outcome of the stove.

Finally, we point out that in the revision we show a model-agnostic signature of MB contributions based on a different idea. We point the reviewer to Fig. 2F-2N and the related discussion (for which the above issue of auto-correlation is also relevant).

We discuss all these issues in SI Note 1.

(3) A similar question arises in figure 4. If the MF system assigns credit to the ghost-chosen object, then reward on the unique room should increase the probability that it will be chosen again on trial $n+1$. Again, $p(\text{repeat})$ does appear to be larger than 0.5 when the common room is non-rewarded (or is it? It is hard to tell from the figure). By contrast, when the common room is rewarded, $p(\text{repeat})$ is much larger than 0.5, both when the unique room is rewarded and when it is not (albeit more so when it is). Could you please clarify?

The repetition probability in this case (unique reward, common non-rewarded) was 0.59, which exceeds 0.5. However, even if it wasn't, the same arguments from the previous point apply. Note, in this case, though that the roles of the unique and common rooms are switched,

since the room that was designated as “unique” (because it reveals the ghost’s choice) was actually the one that is common to both objects on trial n+1.

To avoid confusion between the analyses for Figures 2 and 4, we have changed our terminology for the latter, labelling the rooms as N (Non-informative for inference of the ghost’s nominee) and I (Informative for inference above the ghost’s nominee)

(4) I could not follow the reasoning underlying figure 4B (p.17):

“From the perspective of the MF system, however, if credit was assigned to the ghost-rejected object on trial n, then a unique-reward, as compared to non-reward, should increase the generalization probability.“

Why? In the example shown, the unique brown room disambiguates state uncertainty, it is clear that the ghost-chosen object was the key. If the MB-system is indeed used to guide credit assignment retroactively, the credit for the unique room should be entirely assigned to the choice of “key“, shouldn’t it?

Thanks for raising this important question. We apologise that we failed to stress the critical question of the fraction of trials on which MB occurred and guided MF credit assignment. If this fraction is 1, then the reviewer’s suggestion would be true. If the fraction is less than 1, then since in the absence of MB inference the MF system has no way to distinguish between the chosen and rejected objects, it should therefore assign credit equally to the two.

We now amplify this point in the text related to the findings in the relevant figure. In the section ‘Credit from the Informative outcome is assigned to the ghost-rejected object’ we write:

“This result is striking because it shows that the MF system assigned credit from the Informative outcome to an object that is not related to that outcome. This challenges any notion of perfect MB guidance of MF credit-assignment. However, it is consistent with the possibility that some participants, at least some of the time, do not rely on MB-inference because when MB inference does not occur, or when it fails to guide MF credit-assignment, the MF system has no basis to assign credit unequally to both objects in the selected pair.”

And in the 2nd paragraph of the discussion section we write:

“Note that credit was still assigned to the ghost-rejected object, a finding that is expected if some participants do not retrospectively resolve uncertainty, or if participants resolve it only part of the time.”

Modelling results were all very clear and intuitive.

Thank you very much! We wish to point out that in the revision the modelling section was extended and we hope that the reviewer will find it (even more) useful and clear.

Minor

Please decide to use either „unique“ or „uncommon“ throughout

We changed all instances of “uncommon” to unique. Note also the (nominal) change in terminology in Fig. 4 relative to Fig. 2.

Reviewer #2 (Remarks to the Author):

This paper presents a cleverly designed (though devilishly complicated) experiment designed to reveal how the resolution of state uncertainty can retrospectively assign credit to past actions. First, the authors establish that choices are to some extent controlled by a "model-free" system that learns to select actions based on past reinforcement in particular states. They then establish that the model-free system can assign credit to actions based on later events that resolve state uncertainty. Because this uncertainty resolution can only be accomplished by a "model-based" system, they reason that their findings reflect a cooperative synergy between model-free and model-based systems.

I found the paper overall well-written and compellingly argued. I had to re-read some sections multiple times because the experimental design is quite intricate. I'm not sure if much can be done to improve that aspect of the paper, because the intricacy seems necessary to test for these subtle effects.

We thank the reviewer warmly for a positive, insightful and constructive evaluation of our work. Sorry for the complexity of the design – we know simpler is better but in this instance we couldn't think of a more straightforward approach.

Major comments:

It's somewhat paradoxical that the authors find a correlation between the degree of model-based control and the effect of uncertainty resolution on retrospective credit assignment. On the one hand, I understand the logic of this analysis to be that the retrospective credit assignment is only possible through the operation of a model-based system. On the other hand, if a subject is purely model-based, then they wouldn't show the effect at all. Eyeballing Figure 5C, I think there may actually be some evidence for this: there are a few subjects with a high degree of model-based control who show weaker, or even reversed retrospective credit assignment effects. This might be worth investigating further.

We thank the reviewer for spotting this important observation. However we suggest it is not entirely fair to call this correlation “paradoxical”. It might have been so if adherence to MB control was absolute – with subjects being perfectly MB in their choices with $w_{MB} = 1$. But in this case we could not have measured anything regarding the effect of MB resolution of state uncertainty on a MF system (see also below). If, however, $w_{MB} < 1$, so MB control is only partially effective, then it seems reasonable at least to test the possibility that the more effective it is, the more influence it will have over an internal representational state and in keeping with this the greater the extent to which MF credit assignment might prefer the ghost-selected object (as assessed by the differential learning rates). However, it is important to test this correlation in a way that does not start out from assuming it to be true – and this is precisely what the model does.

One caveat, however, is that measurement noise in estimating MF learning rates and their difference increases as a function of a participant's reliance on the MB system. This happens because the MF learning rates exert a weaker influence on performance as "model-basedness" increases. This is now evident in our revised Fig 6B, to which we add individual error bars. As the reviewer noted, this is also evident in an increased between-participants heterogeneity in the learning rate difference as model-basedness increases. If anything, this increased measurement noise should decrease an ability to detect a significant correlation.

To address this concern more systematically, however, we conducted a follow up analysis. In this analysis, we first fitted to each participant's empirical data our 'non-inference' model, which imposes equal learning rates for the ghost selected and rejected objects. We then generated for each participant based on these fitted parameters 1000 parametric bootstrap datasets by simulating synthetic experimental sessions. Next, we fitted our *full* model (allowing different learning rates for selected and rejected objects) to each synthetic dataset. Finally, we repeated the following 100,000 times: For each participant we selected randomly one of his/her 1000 full-model fitted parameter sets and we took w_{MB} and the learning rate difference (ghost-nominated minus ghost rejected) from this parameter set. We thus obtained 40 "individual" parameters (one per subject) across which we calculated the group correlation (between w_{MB} and the learning rate difference). These 100,000 synthetic correlations comprised a null distribution under a no-correlation hypothesis. The significance value of the empirical correlation was calculated as the proportion of samples in the null distribution that were larger than the observed correlation.

These issues are now addressed in the revised MS in the section "The correlation between MB contributions to choices and specificity of MF learning for the ghost-n object":

"As model-basedness increases, the relative contribution of MF to performance, and hence the influence of MF's learning rates on performance, decreases. Importantly, the full model allowed for an estimation of the extent to which each subject's MF credit assignment prefers the ghost-nominated over the ghost-rejected object ($lr_{ghost-nom} - lr_{ghost-rej}$), controlling for the extent to which that subject relies on MB in his/her choices (w_{MB}). Therefore, we could test for a possible relationship between these variables. Our retrospective-inference theory predicts that MB-inference of the ghost-nominated object, which relies on knowledge of the task's transition structure, directs MF learning towards this inferred object. This suggests that the greater the degree of a subject's model-basedness, the better they can infer the ghost's nominations and hence, the more specific MF-learning might be on uncertainty trials. In support of this hypothesis, we found (Fig. 6B) a positive across-participants correlation ($r = 0.29$, $p = .034$ based on one-sided permutation test) between w_{MB} and $lr_{ghost-nom} - lr_{ghost-rej}$, as quantified by the maximum-likelihood model-parameters (See Fig. S7 for a further control analysis, addressing the concern that a higher learning-rate measurement noise as model-basedness increases can generate a spurious correlation)."

and in Fig. S7 which is appended below for the reviewer's convenience.

Fig. S7, related to Fig. 6B. A bootstrap permutation test for the correlation between model-baseness and preferential MF learning for the ghost-nominated object (see methods). (A-B), two example scatter plots as in Fig 6B, that were obtained by fitting the full model to data that was parametrically bootstrapped from the non-inference model. (C) A null distribution for the no-correlation hypothesis (blue). The data (red) supported a significant correlation.

Another substantive issue is that the modeling is essentially descriptive (different learning rates for different trial types). The authors don't provide a plausible mechanistic account of how resolution of uncertainty contributes to credit assignment. I don't think this is crucial to making the points in their paper, but it might help readers better understand more explicitly what computations would produce these results.

We in fact proposed two potential mechanisms (delayed learning and a form of DYNA) – but we acknowledge that this was buried in the discussion, and evidently not sufficiently amplified. Apologies. We still consider it most appropriate to put these suggestions in the discussion. We have also endeavoured to improve the structuring of the paper to make this aspect of the discussion more salient. First, we conclude the introduction by informing readers that possible mechanisms will be considered in the discussion section (so that interested readers will now anticipate this discussion and know where to find it). We also remind readers of this discussion at the end of the results section “Retrospective MB inference guides MF learning on uncertainty trials”. Second, we revised the structure of the discussion to address these mechanisms earlier. We hope that the reviewer and readers will find the current framing more helpful.

Here is the relevant part from the Discussion:

“...An important question pertains to potential mechanisms that might account for these findings. We consider two, which we discuss in turn, namely delayed learning and a DYNA^{18,28} architecture.

A delayed learning account rests on the idea that an MF-teaching prediction-error signal, based on total trial-reward, is calculated only at the end of uncertainty trials, after inference has already occurred. Notably, our task offered certainty that this uncertainty would be resolved later (i.e., after observing both outcomes). Thus, the MF system could in principle postpone a first-outcome credit assignment and “wait” for a MB inference triggered by observing a second outcome. Once a MB system infers the relevant object it

can presumably increase a MF eligibility trace³⁵ for the inferred object and reduce it for the non-selected object, a process manifest as a higher MF learning rate for the inferred object. Many real-life situations, however, impose greater challenges on learners because it is unknown when, and even whether, uncertainty will be resolved. Indeed, there can often be complex chains of causal inference involving successive, partial, revelations from states and rewards. In such circumstances, postponing credit assignment could be self-defeating. Thus, it may be more beneficial to assign credit in real-time according to one's current belief state, and later on, if and when inferences about past states become available, enact "corrective" credit assignments adjustments. For example, upon taking an action and receiving a reward in an uncertain state, credit could be assigned based on one's belief state. Later, if the original state is inferred, then one can retrospectively deduct credit from the non-inferred state(s) and boost credit for the inferred state. Our findings pave the way for deeper investigations of these important questions and where it would be appealing to exploit tasks in which uncertainties arise, and are resolved, in a more naturalistic manner, gaining realism at the expense of experimental precision. More broadly, we consider much can be learned from studies that address neural processes that support MB-retrospective inference, how such inferences facilitate efficient adaptation, and how they contribute to MB-MF interactions.

Our findings also speak to a rich literature concerning various forms of MB-MF system interactions. In particular, the findings resonate with a DYNA architecture, according to which the MB system trains a MF system by generating offline (i.e., during inter-trial intervals) hypothetical model-based episodes from which an MF system learns as if they were real. Our findings suggest another form of MB-guidance, inference-based guidance whereby upon resolving uncertainty, an MB system indexes appropriate objects for MF credit assignment. One intriguing possibility arises if one integrates our findings with DYNA-like approaches. Consider a case where the MF system assigns credit equally to inferred and non-inferred objects *online* (i.e., during reward administration), but that MB-inference biases the content of *offline* replayed MB-episodes. For example, if the MB system is biased to replay choices of the inferred, as opposed to the non- inferred object (i.e., replay the last trial with its uncertainty resolved), then this would account for enhanced MF-learning for inferred relative to the non-inferred object. Future studies can address this possibility by manipulating variables that affect offline-replay such as cognitive load¹⁸ or perhaps by direct examination of replay that exploits neurophysiological measures."

As the authors know well, the idea that model-based inference guides model-free learning has been pursued in a number of other papers. Some of this literature is already cited (Lak et al., 2017; Starkweather et al., 2018) but not the seminal theoretical work on POMDPs (Kakade & Dayan, 2002; Daw, Courville & Touretzky, 2006; Rao, 2010) nor some of the relevant experimental work (Sarno et al., 2017; Babayan et al., 2018). The citations don't have to be exhaustive, but I thought they were a bit short-shrifted.

We apologise that our citations failed to do justice to the literature. We added citations to all these important references.

Minor comments:

abstract: "an environments" -> "an environment"

Corrected

p. 4: "and calculates the expected rewards" -> "and calculate the expected rewards"

Corrected

p. 4: "an agents'" -> "an agent's"

Corrected

p. 10: I think it would help the reader if the authors unpack what they mean by "canceling out".

Thank you! This was now clarified when we discuss the results pertaining to Fig. 2A-E:

“For example, the calculated MB value of the key on trial $n+1$ is the sum of the MB Q values of the green and brown rooms. Similarly, the calculated MB value of the stove on trial $n+1$ is the sum of the MB Q values for the green and the blue rooms. The MB contribution to choice depends only on the contrast between these calculated key and stove values, which equals the difference between the MB values of the brown and blue rooms. Notably, the value of the green room is not reflected in this contrast (since it is common to both) and hence does not affect MB contributions to choice on trial $n+1$. (Fig 2D)”.

p. 14: "unique, inference allowing, outcome" misplaced commas

Corrected

p. 15: "objected" -> "object"

Corrected

references: the Keramati et al. (2017) citation should be Lak et al. (2017)

Corrected

Reviewer #3 (Remarks to the Author):

In this work, Moran Keramati Dayan Dolan propose a novel theory of goal-directed (model-based; MB) retrospective-inference to resolve uncertainty in the environment that prevail when actions have been taken so as to guide a habitual (model-free; MF) credit-assignment. They moreover experimentally test this theory using a novel task, where there is uncertainty about chosen bandits (i.e. lotteries). They find that when participants' momentary uncertainty about which bandit had generated an outcome was resolved by provision of subsequent

information, participants preferentially assigned credit to the bandit they retrospectively inferred was responsible for an experienced outcome, which suggests evidence for MB retrospective inference.

Overall, the task is well designed, addresses an important question, and the results are very interesting, but would nevertheless deserve a few more control analyses.

We thank the reviewer warmly for a positive, insightful and constructive evaluation of our work.

More importantly, the interpretation of the results (that they would be a hallmark of MB retrospective inference + MF credit assignment) is not yet fully convincing. The authors should control for alternative models such as a pure MF system with eligibility traces, a hybrid MF+MB model without retrospective inference, or a pure MB system learning the values of objects rather than the values of rooms (but still maintaining task structure information linking objects and rooms) which could show the common outcome effect by rewarding the two objects at outcome 1 and then would reinforce the ghost-selected object but not the ghost-rejected object at outcome 2. This model would predict that a third object associated to the same second room would also be reinforced, namely the telephone in the example of Figure 3 because it is also associated to the brown room. I suggest below additional analyses of the present data on other types of trials to help disentangle between alternative models and further illustrate evidence for either MF or MB. Some control analyses for model fitting are also suggested.

Thanks! These important suggestions inspired additional analyses that we include in our revision and which we believe, strengthen our findings. We describe these in detail in our responses to the specific suggestions below.

MAIN ISSUES

I- The result in Figure 4C is very nice and striking. Nevertheless, an important question is whether alternative models could also account for these results. First, the presentation of the task structure in Figure 1B suggests that the objects remain on the screen with the rooms during outcomes 1 and 2. Thus a hybrid MF+MB model without retrospective inference (where the MF reinforces all presented objects while the MB waits for the two outcomes and the two room presentations to know which object is associated to the two rooms and thus only then reinforces this object (rather than reinforcing room values), thus without retrospective-inference) seems a good candidate. Any choice made by this model would be a weighted sum of MF and MB and might thus produce all effects reported here.

Thanks for this interesting suggestion.

To address this issue, we formulated a new family of models that were similar to our existing models but were based on a different principle of MB operation that reflected our understanding of the reviewer's suggestion. In our original models, a MB system learns the values of rooms during reward administration and calculates the values of objects when a choice is asked for (henceforth, a 'room value learning' formulation). In the alternative formulation, an MB system learns the values of objects (not rooms) during reward. However, in contrast to the MF system (and inspired also by sensory preconditioning), the MB system

assigns credit from an observed reward to all the objects that are linked to the room associated with this reward via the task transition structure, and not only to the chosen room (an ‘object value learning’ formulation).

These two formulations are quite similar in spirit, as they both allow an MB system to generalize a reward from an outcome room to non-chosen objects that provide that very same room. However, they differ mathematically (based on an assumption about how reward is generalized) and as such generate different predictions and can be compared with each other. We refer the reviewer to the “Computational Models” method section for details about the object-value learning model. Thus, we had 2 families of models, each consisting of the full model and 4 sub-models of interest (note that in the original MS. we examined only two sub-models. For the revision, we added two new sub models: one pure MB, and the other pure MF).

First, we performed a model comparisons for the ‘object-value learning’ family of models (Fig. S4) and found results that were similar to the results for the ‘room value learning’ models. Namely, each of the 4 sub-models were rejected in favour of the full model. We then continued and compared the room-value learning full model with the object-value learning full model. Here, we used a parametric bootstrap cross-fitting method (PBCM). This comparison showed that the room value learning full model had the upper hand (Fig. S5). Therefore, we retained the description of this model in the main text, but reported predictions of the object-value learning models in Fig. S2-S3.

Importantly, when we re-ran our analyses pertaining to model parameters (Fig. S6), but using parameter values associated with the full object-value learning model (despite its worse fit), the critical conclusions of our study were upheld. We believe that this strengthens our interpretation, since it is evidently robust to model details.

In the main text we present these issues when we formulate the operation of a MB system in the “Model-Free and Model-Based contributions to performance in standard trials”. We write:

“There are various possible MB systems. The most important difference between them concerns whether the MB system learns directly about the rooms, and uses its knowledge of the transition structure to perform an indirect prospective calculation of the values of the objects presented on a trial based on the values of the rooms to which they lead (henceforth, a ‘room value learning’ MB system); or whether it uses knowledge of the transition structure to learn indirectly about the objects, and then uses these for direct evaluation (henceforth, an ‘object value learning’ MB system). While these two formulations of an MB system are similar in that they both allow generalisation of observations about rewards to objects that were not chosen or nominated, they nevertheless differ in their value calculations and generate different experimental predictions.

Until the very end of this section, our presentation relies on the room value learning formulation. This is justified because a model comparison (Fig. S5) revealed that it was superior to an object value learning formulation.”

And we added a new section at the end of the Results section:

“Object value learning MB system: As noted, all the main analyses in this section are based on a room value learning version of MB reasoning. This was motivated by its superior fit to the data. However, in order to support the robustness of our results we repeated our analyses of the influence of MB inference on MF value updating but now assuming that an object value learning version was responsible for the MB values (noting that the MB inference about which object had been nominated by the ghost is unaffected). In this approach we obtained the same conclusions (Fig. S2, S3, S4 and S6).”

The authors could also test whether a pure MF model with eligibility traces (so that any positive outcome received not only reinforces the last selected object but also reinforces a little bit the second to last selected object) could account for this result. This would control if the observed effect is really a hallmark of retrospective inference or whether it could also be explained in terms of MF eligibility traces. The latter would moreover be consistent with the credit assigned during "switch trials" from the unique outcome to the ghost-rejected object. In contrast, the present interpretation for this result, "that an MF system assigns credit to the ghost-rejected object" while the MB system would be responsible for the retrospective inference, does not sound very parsimonious and forces the authors to refer to "MF credit-assignment" throughout the manuscript without really proving it, which I further discuss below.

To address this possibility we formulated, as the reviewer suggested, pure MF models with eligibility trace. We tried two different models, with either accumulating or replacing eligibility traces (as described in Singh & Sutton, 1996). Neither of these two models was successful in predicting the key effects seen in the data. These models are presented in detail in the “Computational Models” method section and their predictions appear in Fig. S1 I-L. We also refer to these issues in the main text at the end of the “computational modelling” results section:

“Finally, because the above models did not include MF eligibility traces³⁴, we examined whether such traces, rather than preferential learning based on retrospective inference, might account for the qualitative “preferential MF credit assignment” patterns presented Fig. 4C-D. We found that models based on eligibility –driven MF learning failed to account for the observed patterns (See methods for full model descriptions; See Fig. S1 I-L for the predictions of these models).”

We hope that the reviewer agrees that these additional controls strengthen confidence in a retrospective-inference account, and thank them for suggesting these further analyses.

The present full model is even more problematic that it does not predict the MF ghost credit-assignment contrast (Figure S2E), in contrast to the experimental data (Figure 4D). In relation to this, I don’t understand why the authors can write that they "validated our analysis-approach using model simulations (see Supplemental Material Fig.S2)" in line 348 while Figure S2E does not reproduce the experimental results with the full model.

We thank the reviewer for raising this point. We dropped the ‘validation’ terminology and instead we point out more clearly the goal of these model simulations. For all the models that

we simulated the only account that generated the effects in both Fig. 4C and 4D was a MF treatment with different learning rates for ghost-selected and ghost-rejected objects (so demanding MB inference). We fully acknowledge that this MF account nevertheless fits the whole collection of choices less well than our full model, even though the latter does not account for the contrast in Fig. 4D. This suggests that there is room for refinement. However, we suggest that the generally converging evidence from model-agnostic analyses, formal model comparisons (fig. 5) and analyses based on estimated parameters (Fig. 6) provide a compelling convergence of support for our conclusions.

Note that we extended supplemental figure S1 include additional models including, for example, the pure MF eligibility trace models suggested by the reviewer; See also simulations of the object-value learning models in Fig. S3).

II- As I understand Figure 2, it is intended to illustrate some predictions of the model and then compare to subjects' behavior. In the first place, it could be merged with Figure S1, showing (A) the MF-contribution trials, (B-D) model predictions and then (E) the subjects' behavior. Then I think it would also be great to add a second line to Figure 2 (panels F-J) to highlight, symmetrically, some MB-contribution trials: cases where the same object is not presented at trial $n+1$, but rather an object that shares one common room (in this example, the stove or the phone) against an object that shares no room (the light bulb), and show how frequently MF and MB chose the latter as a function of C-Non/C-Rew and whether the two different rooms (D1 and D2) of the light bulb were both rewarded, both unrewarded, or only one rewarded. These additional panels would show a significant difference between blue (C-Non) and red (C-Rew) curves for MB but not for MF, which I think would further clarify different predictions of the two systems.

Thanks for this helpful suggestion. In the revision we include a novel marker of MB processing that focuses on an analysis of trials as the reviewer suggested (See the results focusing on Fig. 2F-N). However, we point out that (unfortunately) things are somewhat more complicated, because a pure MF system can also predict a positive effect for the common outcome on such trials (See Fig. 2I).

The explanation for this is subtle, since it depends on the autocorrelation in the reward statistics (see also the explanation of fig 2A in response to reviewer 1). We therefore provide an intuition by focusing on the example that the reviewer outlined and that is now presented in Fig. 2F. This involves a choice of a key on trial n which is followed with a choice of stove vs. bulb on trial $n+1$. The question is why would the tendency of an MF system to choose the stove (which shares the green outcome with the key) be higher following a reward (vs. non-reward) for the green room on trial n .

The key to understanding this phenomenon is the observation that while the trial- n green-outcome event (reward or non-reward) cannot exert a direct causal influence on the MF value of the stove on trial $n+1$, it is, nevertheless, correlated with the value of the stove on trial $n+1$. Indeed, because the rewards for each room were generated from a random walk, the probability-reward time series for each room is auto-correlated. This means that each room will alternate between temporally extended epochs when it is either better or worse than its average. When the green room is rewarded vs. unrewarded on trial n , it is more likely that the green room is currently in a good epoch. Consequently, the stove was more likely to earn a

reward from the green room when it was recently chosen *prior to trial n*. In sum, a green outcome for the key on trial n (non-reward vs. reward), is positively correlated with the MF value of the stove on trial n+1. We stress that this correlation is not due to direct causal influences (trial n \rightarrow trial n+1) but due to the temporal autocorrelation of the reward time series.

Importantly, this reasoning also suggests that by controlling for the green reward probability, one can wash away the MF contributions to the effect of the trial n green outcome on (trial n+1) choice and hence isolate a pure MB contribution (Fig. 2H-I). As we show, this is indeed the case. Indeed, a pure MF system predicted that when the green reward probability is added to the regression it will show a positive effect on the generalization choice probability, and the effect of the trial n green outcome (reward or non-reward) would vanish. A MB system, on the other hand, predicts that both the last green outcome (from trial n) and the reward probability will have a positive effect on the trial n+1 stove-choice probability. We note that an MB contribution predicts an effect for the reward probability above and beyond the effect of the trial n green outcome, because this reward probability correlates with green-reward events prior to trial n and these past reward events affect the MB Q value of the stove on trial n+1. Critically, empirical data showed a significantly positive regression-coefficient for the actual green outcome above and beyond the effect of the green reward-probability, implicating an MB contribution. We present these issues in the text related to Fig. 2.

“Turning next to a MB contribution, consider a trial-n+1 which excludes the trial-n chosen object (e.g., the key; Fig. 2F) from the choice set. In this case, the trial-n chosen object shares a Common room (e.g., green) with only one of the trial-n+1 offered objects (e.g., stove), whose choice we label a “generalization”. Additionally, the trial n-chosen object shares no outcome with the other trial-n+1 offered object (e.g., light bulb). We examined whether the probability of generalizing the choice depended on the Common outcome on trial-n (Fig. 2G). A MB contribution (Fig. 2H) predicts a higher probability to generalize the choice when the Common-room was rewarded on trial-n, as compared to non-rewarded, because this reward increases the calculated Q-values of all objects (including the stove) that open that room.

Considering the MF system (Fig. 2I), trial-n reward-events cannot causally affect choices on trial-n+1 because learning on trial-n was restricted to the chosen object which is not present on trial n+1. The predictions for the MF system, however, are somewhat complicated by the fact that a Common green outcome on trial n (reward vs. non-reward) is positively *correlated* with the MF Q-value of the stove on trial-n+1. To understand this correlation, note that the reward probability time series for each room is auto-correlated since it follows a random walk. This means coarsely that the green room’s reward probability time series alternates between temporally extended epochs during which the green room is better or worse than its average in terms of a reward probability. When the green room is rewarded vs. unrewarded on trial-n, it is more likely that the green room is currently spanning one of its better epochs. Importantly, this also means that the stove was more likely to earn a reward from the green room when it had recently been chosen *prior to trial n*. Thus, a Common-room reward for the key on trial n, is positively correlated with the MF value of the stove on trial n+1. It follows that an MF contribution predicts a higher

generalization probability when the Common room is rewarded as compared to non-rewarded. Critically, because this MF prediction is mediated by the reward probability of the Common room (i.e., how good the green room is in the current epoch), a control for this probability washed away a MF contribution to the effect of the Common trial-n outcome on choice generalisation (Fig. 2M) generalization, hence implicating the contribution of an MB system (Fig. 2L).

A logistic mixed effects model showed (Fig. 2K) a positive main effect for the Common-outcome ($b= 0.40$, $t(3225)= 3.328$, $p= 9e-4$) on choice-generalization, supporting an MB contribution to bandit choices. Additionally, we found a significant main effect for the Common outcome's reward probability ($b= 1.94$, $t(3225)= 7.08$, $p=2e-12$) as predicted by both systems (Fig. 2L, 2M). We found no interaction between the Common trial-n outcome and the Common room's reward probability ($b= -.75$, $t(3225)= -1.76$, $p= .079$). Note that unlike our first analysis which pertained to an MF contribution, the current analysis did not control for the effect of the Unique room (e.g., brown), because it was an outcome of neither choice object on trial-n+1. Hence, this room's outcome was expected to exert no influence on choice generalization from the perspective of either MB or MF. Indeed, when we added the Unique room trial-n outcome to the mixed effect model, none of the effects involving the Unique room outcome were significant (all $p>.05$), and the effects of the Common room's outcome and Common reward probability remained significantly positive (both $p<.001$) with no interaction. Consequently, this room was not considered in our analyses pertaining to a MB's contribution to performance."

We should point out that we did not include in this analysis the rewards earned from the stove on the last time it was chosen, as the reviewer suggested, because this is non-essential for showing a MB contribution and it would just complicate the analysis.

In the revision we extended Fig. 2 as the reviewer suggested to include both data and model predictions.

III- I am not totally convinced by the authors in lines 356-357 that "MB contribution predicts no effect of the trial n common outcome due to cancelling out", because one could think of a slightly different MB model where the model where the model learns the value of objects rather than the value of rooms, still using task structure information (as written above). The model would make different predictions than that of the present model. So the point raised by the authors does not seem as a general MB prediction, but rather specific to the MB that they are using.

The reviewer is correct. As we explained in our response to point I above, we formulated an alternative "object-value learning" MB (though we kept the descriptions of the 'room-value learning' models in the main text). However, we also show the prediction of a pure MB system based on the alternative object-value learning formulation (Fig. S1; See also Fig. S3). We found that this alternative pure MB model cannot account for empirical data patterns.

IV- To further disentangle MF and MB predictions, and similarly to the suggestions above

for Figure 2, I would suggest adding another type of trials in Figure 4: uncertainty trials n followed by trials $n+1$ with a choice between an object associated to none of the rooms and an object associated to the unique room but not the common one (light bulb against telephone in the example of Figure 4). A preference for the latter would indicate that during outcome 2, credit was not assigned only to the object which is associated to the two rooms presented at outcome 1 and outcome 2 (the key in the example of Figure 4), but rather to all objects associated to the second room (key and telephone are both associated with the brown room).

Thanks for this suggestion. As described in the methods section, and now also in the caption of Fig. 4, in our design, each trial that followed an uncertain ghost, trial was either a repeat, a switch or a clash trials. Therefore, we cannot pursue the suggested analysis. However, as described in our response to point II above, we do now present a novel marker of an MB contribution, which shows that an MB system generalized credit from an outcome to a non-chosen object that yields that very same outcome.

V- A word about the terminology. From the end of the introduction and throughout the manuscript, the authors repeatedly employ the expressions "MF credit-assignment", "MF assignment of credit", "MF learning". First, this sounds a bit strange when it is associated with MB inference or with the MB system in the same sentence (for instance lines 96-97) because the reader does not yet know that the authors intend to later present results to specifically address MF learning, and thus does not understand at this stage why the credit assignment should necessarily be MF rather than simply be a later preference of the MF system towards the inferred state. One may think of the DYNA architecture by Richard Sutton, which is cited by the authors. But it is only at the end of the manuscript that the authors really discuss this.

We thank the reviewer for these suggestions. We revised the introduction to reflect more precisely our interest in an interaction between the MB and MF systems. We also highlight the fact that our paradigm allowed a dissociation between MB and MF learning. We also anticipate a connection with DYNA in the end of the introduction. However, we preferred to defer the detailed discussion of this point to the discussion section since it is not something that we test explicitly in this task.

Second, the authors continue to use the same expressions when presenting results that focus on MB retrospective-inference, while at these other moments it does not really matter whether the credit assignment is done in a MF way or a MB way. For instance, lines 149-152, lines 228-229, lines 247-248, lines 261-263, line 309. So, to facilitate reading, I would suggest to replace all occurrences of "MF learning", "MF credit-assignment", "MF assignment of credit" by "learning", "credit-assignment", "assignment of credit", respectively.

We would actually like to stress the interaction between a MB inference and implications for MF credit assignment. So we prefer to maintain this line of narration. We hope that our editing (see our response to the previous paragraph) and our additional controls, as the reviewer suggested, makes this point clearer.

Finally, and still with respect to the same issue, the reinforcement arrow for the key in Figure 3 is thicker than that of the stove not only after Outcome 2 but also after Outcome 1, while at this stage of the reading, the reader does not know that the subjects showed this effect (Figure

4) and rather think that the arrows should be equally thick because the two objects should be equally rewarded. A note in the figure legend would be useful.

Thanks for the suggestion. This is now clarified in the figure caption.

VI- Individual differences. In lines 403 and 406 respectively, 14 and 10 participants respectively are both minorities of the total of 40 participants. Could the authors be more explicit about which alternative models win for the other participants at the individual-level?

Thanks for this question. As we now clarify, our model comparisons are based on the classical-statistics hypothesis-testing approach. In each of our comparisons, a constrained sub-model serves as a null hypothesis which we are trying to reject, whereas the full model serves as the alternative H1 hypothesis. Thus, if a null sub-model is rejected for 10 participants, then for the rest 30 individuals, the null model cannot be rejected (and in that sense it “wins”).

Although these two sub-models were rejected for a minority of the individuals, we find this minority is of substantial size. We note three facts. First, tasks of the sort we ran here are not optimally powered to detect individual effects. This would require much more extensive testing. Second, based on a null hypothesis one would expect to obtain only about 2 (5% of 40) rejections of the null. Rejecting 10 subjects would be extremely unlikely. Finally, when we applied the Benjamini-Hochberg method to control for the rate of false discoveries the models were rejected for 10 (non-inference) or 4 (no learning for ghost-rejected) participants. We now present this issue in the computation modelling results section:

“We note that although the ‘non-inference’ and the ‘no learning for ghost rejected’ models were each rejected for a minority of the participants (10 and 12 participants, respectively), the size of these minorities are nevertheless substantial considering that our task was not optimally powered to detect individual participant effects, and given significance testing at $p=.05$ should yield on average 2 rejections (out of 40 participants) when the null hypothesis holds for all participants. Finally, applying the Benjamini-Hochberg procedure³⁴ to control for the false discovery rate, the ‘non-inference’ and the ‘no learning for ghost rejected’ models were rejected for 10 and 4 individuals, respectively”

VII- Model fitting. Could the authors verify in which percentage of the 20 starting points per model and per participant the gradient descent performed by `fmincon` converged to the same point? If this percentage is small, the authors may consider to increase the number of starting points in order to increase the chances of converging to the global optimum.

Thanks for this suggestion. Convergence properties were usually very good for most participants and for all models. For example, for the full model, an average of 55% of fitting attempts converged to the same point (for the sub-models with fewer parameters this percentage was even higher). Nevertheless, to be on a safer side we increased the number of starting points by a 10-fold to 200. This had only slight influences on the fitted parameters and no influence on our conclusions.

SECONDARY ISSUES

Does this really make sense: "vertical counterpart from the non-selected pair (this is the object that the ghost would have chosen had the participant chosen the other pair)"? How can the subject know what the ghost would have chosen from the non-selected pair? Why would one object be more likely chosen by the ghost within that non-selected pair?

Thanks. Sorry for the lack of clarity in our description. We have now tried to describe the trials analysed in Fig. 4A and 4B more precisely.

Fig. 4A focuses on trials that presented the ghost-selected object alongside the object from the previously non-chosen pair, which shared the previously inference-allowing outcome with the ghost selected object. Fig. 4B focuses on trials that presented the ghost-rejected object alongside that object from the rejected pair that shares a room-outcome with the ghost rejected object (this shared room was not experienced on the uncertainty trial).

Please keep a consistent terminology, for instance "unique outcome" rather than "second-outcome" in Figure S2 legend.

Done. Please also note that to avoid confusion we changed the terminology related to Fig. 4 to "Informative" and "Non-informative" room.

In line 381, the authors should have written "ghost_rej" as subscript of the third MF learning rate.

Corrected

In lines 674-677 the title of the article is repeated twice. Moreover, the first author is Armin Lak, not Mehdi Keramati. Finally, this reference is repeated in lines 740-742.

Corrected

REVIEWERS' COMMENTS:

Reviewer #1 (Remarks to the Author):

Thank you for the detailed revisions of the manuscript and the helpful clarifications and additional explanations provided in your response letter. I think you did a great job at explaining intricate predictions and behavioural findings with great clarity. Congratulations!

Reviewer #2 (Remarks to the Author):

The authors have addressed all of my concerns.

Reviewer #3 (Remarks to the Author):

The authors have done a great job in performing additional analyses and improving the manuscript.

An important issue remains: in Fig. S2, the data is apparently not reported accurately. In panel K, coefficient beta should also be significant for the trial-n Common room's outcome (unrewarded vs. rewarded; Rew). It is indeed indicated as significant in panel K of Fig. 2 in the main manuscript.

We are grateful to the reviewers for their positive evaluation of our paper. We have revised the MS. to address the remaining issue raised by R3 as detailed below.

Reviewer #3 (Remarks to the Author):

An important issue remains: in Fig. S2, the data is apparently not reported accurately. In panel K, coefficient beta should also be significant for the trial-n Common room's outcome (unrewarded vs. rewarded; Rew). It is indeed indicated as significant in panel K of Fig. 2 in the main manuscript.

We thank the reviewer for spotting this mistake. We apologize. When we copied the main figure panel 2K into the supplement figure S2, we mistakenly dropped the significance indicator (***) for the "Rew coefficient".

Importantly, we confirm that the results reported in main Fig 2K are correct and we corrected the corresponding supplementary figure 2-K (it is now identical to Fig. 2K). Incidentally, there was a similar mistake in Supplementary figure 2-M (we dropped the significance indicator of "Prob" repressor when we copied Fig. 2M). We corrected this as well.